# A random matrix analysis of random Fourier features: beyond the Gaussian kernel, a precise phase transition, and the corresponding double descent

**Zhenyu Liao**
ICSI and Department of Statistics
University of California, Berkeley, USA
zhenyu.liao@berkeley.edu

**Romain Couillet**
G-STATS Data Science Chair, GIPSA-lab
University Grenobles-Alpes, France
romain.couillet@gipsa-lab.grenoble-inp.fr

**Michael W. Mahoney**
ICSI and Department of Statistics
University of California, Berkeley, USA
mmahoney@stat.berkeley.edu

## Abstract

This article characterizes the exact asymptotics of random Fourier feature (RFF) regression, in the realistic setting where the number of data samples $n$, their dimension $p$, and the dimension of feature space $N$ are all large and comparable. In this regime, the random RFF Gram matrix no longer converges to the well-known limiting Gaussian kernel matrix (as it does when $N \to \infty$ alone), but it still has a tractable behavior that is captured by our analysis. This analysis also provides accurate estimates of training and test regression errors for large $n, p, N$. Based on these estimates, a precise characterization of two qualitatively different phases of learning, including the phase transition between them, is provided; and the corresponding double descent test error curve is derived from this phase transition behavior. These results do not depend on strong assumptions on the data distribution, and they perfectly match empirical results on real-world data sets.

## 1 Introduction

For a machine learning system having $N$ parameters, trained on a data set of size $n$, asymptotic analysis as used in classical statistical learning theory typically either focuses on the (statistical) population $n \to \infty$ limit, for $N$ fixed, or the over-parameterized $N \to \infty$ limit, for a given $n$. These two settings are technically more convenient to work with, yet less practical, as they essentially assume that one of the two dimensions is negligibly small compared to the other, and this is rarely the case in practice. Indeed, with a factor of 2 or 10 more data, one typically works with a more complex model. This has been highlighted perhaps most prominently in recent work on neural network models, in which the model complexity and data size increase together. For this reason, the *double asymptotic* regime where $n, N \to \infty$, with $N/n \to c$, a constant, is a particularly interesting (and likely more realistic) limit, despite being technically more challenging [48, 51, 21, 15, 37, 32, 5]. In particular, working in this regime allows for a finer quantitative assessment of machine learning systems, as

a function of their *relative* complexity $N/n$, as well as for a precise description of the under- to over-parameterized "phase transition" (that does not appear in the $N \to \infty$ alone analysis). This transition is largely hidden in the usual style of statistical learning theory [49], but it is well-known in the statistical mechanics approach to learning theory [48, 51, 21, 15], and empirical signatures of it have received attention recently under the name "double descent" phenomena [1, 7].

This article considers the asymptotics of random Fourier features [43], and more generally random feature maps, which may be viewed also as a single-hidden-layer neural network model, in this limit. More precisely, let $\mathbf{X} = [\mathbf{x}_1, \dots, \mathbf{x}_n] \in \mathbb{R}^{p \times n}$ denote the data matrix of size $n$ with data vectors $\mathbf{x}_i \in \mathbb{R}^p$ as column vectors. The random feature matrix $\mathbf{\Sigma_X}$ of $\mathbf{X}$ is generated by pre-multiplying some random matrix $\mathbf{W} \in \mathbb{R}^{N \times p}$ having i.i.d. entries and then passing through some *entry-wise* nonlinear function $\sigma(\cdot)$, i.e., $\mathbf{\Sigma_X} \equiv \sigma(\mathbf{WX}) \in \mathbb{R}^{N \times n}$. Commonly used random feature techniques such as random Fourier features (RFFs) [43] and homogeneous kernel maps [50], however, rarely involve a single nonlinearity. The popular RFF maps are built with cosine and sine nonlinearities, so that $\mathbf{\Sigma_X} \in \mathbb{R}^{2N \times n}$ is obtained by cascading the random features of both, i.e., $\mathbf{\Sigma_X}^\mathsf{T} \equiv [\cos(\mathbf{WX})^\mathsf{T}, \ \sin(\mathbf{WX})^\mathsf{T}]$. Note that, by combining both nonlinearities, RFFs generated from $\mathbf{W} \in \mathbb{R}^{N \times p}$ are of dimension $2N$.

The large $N$ asymptotics of random feature maps is closely related to their limiting kernel matrices $\mathbf{K_X}$. In the case of RFF, it was shown in [43] that *entry-wise* the Gram matrix $\mathbf{\Sigma_X}^\mathsf{T}\mathbf{\Sigma_X}/N$ converges to the Gaussian kernel matrix $\mathbf{K_X} \equiv \{\exp(-\|\mathbf{x}_i - \mathbf{x}_j\|^2/2)\}_{i,j=1}^n$, as $N \to \infty$. This follows from $\frac{1}{N}[\mathbf{\Sigma_X}^\mathsf{T}\mathbf{\Sigma_X}]_{ij} = \frac{1}{N}\sum_{t=1}^N \cos(\mathbf{x}_i^\mathsf{T}\mathbf{w}_t)\cos(\mathbf{w}_t^\mathsf{T}\mathbf{x}_j) + \sin(\mathbf{x}_i^\mathsf{T}\mathbf{w}_t)\sin(\mathbf{w}_t^\mathsf{T}\mathbf{x}_j)$, for $\mathbf{w}_t$ independent Gaussian random vectors, so that by the strong law of large numbers, for fixed $n, p$, $[\mathbf{\Sigma_X}^\mathsf{T}\mathbf{\Sigma_X}/N]_{ij}$ goes to its expectation (with respect to $\mathbf{w} \sim \mathcal{N}(\mathbf{0}, \mathbf{I}_p)$) almost surely as $N \to \infty$, i.e.,

$$[\mathbf{\Sigma_X}^\mathsf{T}\mathbf{\Sigma_X}/N]_{ij} \xrightarrow{a.s.} \mathbb{E}_\mathbf{w}\left[\cos(\mathbf{x}_i^\mathsf{T}\mathbf{w})\cos(\mathbf{w}^\mathsf{T}\mathbf{x}_j) + \sin(\mathbf{x}_i^\mathsf{T}\mathbf{w})\sin(\mathbf{w}^\mathsf{T}\mathbf{x}_j)\right] \equiv \mathbf{K}_{\cos} + \mathbf{K}_{\sin}, \quad (1)$$

with

$$\mathbf{K}_{\cos} + \mathbf{K}_{\sin} \equiv e^{-\frac{1}{2}(\|\mathbf{x}_i\|^2 + \|\mathbf{x}_j\|^2)}\left(\cosh(\mathbf{x}_i^\mathsf{T}\mathbf{x}_j) + \sinh(\mathbf{x}_i^\mathsf{T}\mathbf{x}_j)\right) = e^{-\frac{1}{2}(\|\mathbf{x}_i - \mathbf{x}_j\|^2)} \equiv [\mathbf{K_X}]_{ij}. \quad (2)$$

While this result holds in the $N \to \infty$ limit, recent advances in random matrix theory [30, 27] suggest that, in the more practical setting where $N$ is not much larger than $n, p$ and $n, p, N \to \infty$ at the same pace, the situation is more subtle. In particular, the above entry-wise convergence remains valid, but the convergence $\|\mathbf{\Sigma_X}^\mathsf{T}\mathbf{\Sigma_X}/N - \mathbf{K_X}\| \to 0$ no longer holds in spectral norm, due to the factor $n$, now large, in the norm inequality $\|\mathbf{A}\|_\infty \leq \|\mathbf{A}\| \leq n\|\mathbf{A}\|_\infty$ for $\mathbf{A} \in \mathbb{R}^{n \times n}$ and $\|\mathbf{A}\|_\infty \equiv \max_{ij}|\mathbf{A}_{ij}|$. This implies that, in the large $n, p, N$ regime, the assessment of the behavior of $\mathbf{\Sigma_X}^\mathsf{T}\mathbf{\Sigma_X}/N$ via $\mathbf{K_X}$ may result in a spectral norm error that blows up. As a consequence, for various machine learning algorithms [10], the performance guarantee offered by the limiting Gaussian kernel is less likely to agree with empirical observations in real-world large-scale problems, when $n, p$ are large.[1]

## 1.1 Our Main Contributions

We consider the RFF model in the more realistic large $n, p, N$ limit. While, in this setting, the RFF empirical Gram matrix does *not* converge to the Gaussian kernel matrix, we can characterize its behavior as $n, p, N \to \infty$ and provide *asymptotic performance guarantees* for RFF on large-scale problems. We also identify a phase transition as a function of the ratio $N/n$, including the corresponding double descent phenomenon. In more detail, our contributions are the following.

1.We provide a *precise* characterization of the asymptotics of the RFF empirical Gram matrix, in the large $n, p, N$ limit (Theorem 1). This is accomplished by constructing a deterministic equivalent for the resolvent of the RFF Gram matrix. Based on this, the behavior of the RFF model is (asymptotically) accessible through a fixed-point equation, that can be interpreted in terms of an angle-like correction induced by the non-trivial large $n, p, N$ limit (relative to the $N \to \infty$ alone limit).

2.We derive the asymptotic training and test mean squared errors (MSEs) of RFF ridge regression, as a function of the ratio $N/n$, regularization penalty $\lambda$, training as well as test sets (Theorem 2 and 3, respectively). We identify precisely the under- to over-parameterization phase transition, as a function

of the relative model complexity $N/n$; we prove the existence of a "singular" peak of test error at the $N/n = 1/2$ boundary; and we characterize the corresponding *double descent* behavior. Importantly, our results are valid *with almost no specific assumption* on the data distribution. This is a significant improvement over existing double descent analyses, which fundamentally rely on the knowledge of the data distribution (often assumed to be multivariate Gaussian for simplicity) [20, 36].

3.We provide a detailed empirical evaluation of our theoretical results, demonstrating that the theory closely matches empirical results on a range of real-world data sets (Section 3 and Section F in the supplementary material). This includes the correction due to the large $n, p, N$ setting, sharp transitions (as a function of $N/n$) in angle-like quantities, and the corresponding double descent test curves. This also includes an evaluation of the impact of training-test similarity and the effect of different data sets, thus confirming, as stated in 2., that (unlike in prior work) the phase transition and double descent hold with almost no specific assumption on the data distribution.

## 1.2 Related Work

Here, we provide a brief review of related previous efforts.

**Random features and limiting kernels.** In most RFF work [44, 4, 3, 45], non-asymptotic bounds are given, on the number of random features $N$ needed for a predefined approximation error of a given kernel matrix with fixed $n, p$. A more recent line of work [2, 14, 22, 9] has focused on the over-parameterized $N \to \infty$ limit of large neural networks by studying the corresponding *neural tangent kernels*. Here, we position ourselves in the more practical regime where $n, p, N$ are all large and comparable, and provide *asymptotic performance guarantees* that better fit large-scale problems.

**Random matrix theory.** From a random matrix theory perspective, nonlinear Gram matrices of the type $\mathbf{\Sigma}_{\mathbf{X}}^{\mathsf{T}}\mathbf{\Sigma}_{\mathbf{X}}$ have recently received an unprecedented research interests, due to their close connection to neural networks [41, 39, 8, 38], with a particular focus on the associated eigenvalue distribution. Here we propose a deterministic equivalent [11, 19] analysis for the resolvent matrix that provides access, not only to the eigenvalue distribution, but also to the regression error of central interest in this article. While most existing deterministic equivalent analyses are performed on linear models, here we focus on the *nonlinear* RFF model. From a technical perspective, the most relevant work is [30, 36]. We improve their results by considering *generic* data model on the popular RFF model.

**Statistical mechanics of learning.** A long history of connections between statistical mechanics and machine learning models (such as neural networks) exists, including a range of techniques to establish generalization bounds [48, 51, 21, 15], and recently there has been renewed interest [32, 34, 33, 35, 5]. Their relevance to our results lies in the use of the thermodynamic limit (akin to the large $n, p, N$ limit), rather than the classical limits more commonly used in statistical learning theory, where uniform convergence bounds and related techniques can be applied.

**Double descent in large-scale learning systems.** The large $n, N$ asymptotics of statistical models has received considerable research interests in the machine learning community [40, 20], resulting in a (somehow) counterintuitive phenomenon referred to as the "double descent." Instead of focusing on different "phases of learning" [48, 51, 21, 15, 32], the "double descent" phenomenon focuses on an empirical manifestation of the phase boundary and refers to the empirical observations of the test error curve as a function of the model complexity, which differs from the usual textbook description of the bias-variance tradeoff [1, 26, 7, 17]. Theoretical investigation into this phenomenon mainly focuses on various regression models [13, 6, 12, 25, 20, 36]. In most cases, quite specific (and rather strong) assumptions are imposed on the input data distribution. In this respect, our work extends the analysis in [36] to handle the RFF model and its phase structure *on real-world data sets*.

## 1.3 Notations and Organization of the Paper

Throughout this article, we follow the convention of denoting scalars by lowercase, vectors by lowercase boldface, and matrices by uppercase boldface letters. In addition, the notation $(\cdot)^{\mathsf{T}}$ denotes the transpose operator; the norm $\|\cdot\|$ is the Euclidean norm for vectors and the spectral or operator norm for matrices; and $\xrightarrow{a.s.}$ stands for almost sure convergence of random variables.

Our main results on the asymptotic training and test MSEs of RFF ridge regression are presented in Section 2, with proofs deferred to the Appendix. In Section 3, we provide detailed empirical evaluations of our main results, as well as discussions on the corresponding phase transition behavior and the double descent test curve. Concluding remarks are placed in Section 4. For more detailed discussions and empirical evaluations, we refer the readers to an extended version of this article [28].

## 2    Main Technical Results

In this section, we present our main theoretical results. To investigate the large $n, p, N$ asymptotics of the RFF model, we shall technically position ourselves under the following assumption.

**Assumption 1.** *As $n \to \infty$, we have*

1. $0 < \liminf_n \min\{\frac{p}{n}, \frac{N}{n}\} \leq \limsup_n \max\{\frac{p}{n}, \frac{N}{n}\} < \infty$; *or, practically speaking, the ratios $p/n$ and $N/n$ are only moderately large or moderately small.*

2. $\limsup_n \|\mathbf{X}\| < \infty$ *and* $\limsup_n \|\mathbf{y}\|_\infty < \infty$, *i.e., they are normalized with respect to $n$.*

Under Assumption 1, we consider the RFF regression model. For training data $\mathbf{X} \in \mathbb{R}^{p \times n}$ of size $n$, the associated random Fourier features, $\boldsymbol{\Sigma}_\mathbf{X} \in \mathbb{R}^{2N \times n}$, are obtained by computing $\mathbf{WX} \in \mathbb{R}^{N \times n}$, for standard Gaussian random matrix $\mathbf{W} \in \mathbb{R}^{N \times p}$, and then applying entry-wise cosine and sine nonlinearities on $\mathbf{WX}$, i.e., $\boldsymbol{\Sigma}_\mathbf{X}^\mathsf{T} = [\cos(\mathbf{WX})^\mathsf{T}, \ \sin(\mathbf{WX})^\mathsf{T}]$ with $\mathbf{W}_{ij} \sim \mathcal{N}(0, 1)$. Given this setup, the RFF ridge regressor $\boldsymbol{\beta} \in \mathbb{R}^{2N}$ is given by, for $\lambda \geq 0$,

$$\boldsymbol{\beta} \equiv \frac{1}{n}\boldsymbol{\Sigma}_\mathbf{X} \left( \frac{1}{n}\boldsymbol{\Sigma}_\mathbf{X}^\mathsf{T}\boldsymbol{\Sigma}_\mathbf{X} + \lambda\mathbf{I}_n \right)^{-1} \mathbf{y} \cdot 1_{2N>n} + \left( \frac{1}{n}\boldsymbol{\Sigma}_\mathbf{X}\boldsymbol{\Sigma}_\mathbf{X}^\mathsf{T} + \lambda\mathbf{I}_{2N} \right)^{-1} \frac{1}{n}\boldsymbol{\Sigma}_\mathbf{X}\mathbf{y} \cdot 1_{2N<n}. \quad (3)$$

The two forms of $\boldsymbol{\beta}$ in (3) are equivalent for any $\lambda > 0$ and minimize the (ridge-regularized) squared loss $\frac{1}{n}\|\mathbf{y} - \boldsymbol{\Sigma}_\mathbf{X}^\mathsf{T}\boldsymbol{\beta}\|^2 + \lambda\|\boldsymbol{\beta}\|^2$ on the training set $(\mathbf{X}, \mathbf{y})$. Our objective is to characterize the large $n, p, N$ asymptotics of both the *training MSE*, $E_{\text{train}}$, and the *test MSE*, $E_{\text{test}}$, defined as

$$E_{\text{train}} = \frac{1}{n}\|\mathbf{y} - \boldsymbol{\Sigma}_\mathbf{X}^\mathsf{T}\boldsymbol{\beta}\|^2, \quad E_{\text{test}} = \frac{1}{\hat{n}}\|\hat{\mathbf{y}} - \boldsymbol{\Sigma}_{\hat{\mathbf{X}}}^\mathsf{T}\boldsymbol{\beta}\|^2, \quad (4)$$

with $\boldsymbol{\Sigma}_{\hat{\mathbf{X}}}^\mathsf{T} \equiv [\cos(\mathbf{W}\hat{\mathbf{X}})^\mathsf{T}, \ \sin(\mathbf{W}\hat{\mathbf{X}})^\mathsf{T}] \in \mathbb{R}^{\hat{n} \times 2N}$ on a test set $(\hat{\mathbf{X}}, \hat{\mathbf{y}})$ of size $\hat{n}$.

### 2.1    Asymptotic Deterministic Equivalent

To start, we observe that the training MSE, $E_{\text{train}}$, in (4), can be written as $E_{\text{train}} = \frac{\lambda^2}{n}\|\mathbf{Q}(\lambda)\mathbf{y}\|^2 = -\frac{\lambda^2}{n}\mathbf{y}^\mathsf{T}\partial\mathbf{Q}(\lambda)\mathbf{y}/\partial\lambda$, which depends on the quadratic form $\mathbf{y}^\mathsf{T}\mathbf{Q}(\lambda)\mathbf{y}$ of

$$\mathbf{Q}(\lambda) \equiv \left( \frac{1}{n}\boldsymbol{\Sigma}_\mathbf{X}^\mathsf{T}\boldsymbol{\Sigma}_\mathbf{X} + \lambda\mathbf{I}_n \right)^{-1} \in \mathbb{R}^{n \times n}, \quad (5)$$

the so-called *resolvent* of $\frac{1}{n}\boldsymbol{\Sigma}_\mathbf{X}^\mathsf{T}\boldsymbol{\Sigma}_\mathbf{X}$ (also denoted $\mathbf{Q}$ when there is no ambiguity) with $\lambda > 0$.

In order to assess the asymptotic training MSE, it thus suffices to find a deterministic equivalent for $\mathbf{Q}(\lambda)$ (i.e., a *deterministic* matrix that captures the asymptotic behavior of the latter). One possibility is the expectation $\mathbb{E}_\mathbf{W}[\mathbf{Q}(\lambda)]$. Informally, if the training MSE $E_{\text{train}}$ (that is random due to random $\mathbf{W}$) is "close to" some deterministic quantity $\bar{E}_{\text{train}}$, in the large $n, p, N$ limit, then $\bar{E}_{\text{train}}$ must have the same limit as $\mathbb{E}_\mathbf{W}[E_{\text{train}}] = -\frac{\lambda^2}{n}\partial\mathbf{y}^\mathsf{T}\mathbb{E}_\mathbf{W}[\mathbf{Q}(\lambda)]\mathbf{y}/\partial\lambda$ for $n, p, N \to \infty$. However, $\mathbb{E}_\mathbf{W}[\mathbf{Q}]$ involves integration (with no closed-form due to the matrix inverse), and it is not a convenient quantity with which to work. Our objective is to find an asymptotic "alternative" for $\mathbb{E}_\mathbf{W}[\mathbf{Q}]$ that is (i) close to $\mathbb{E}_\mathbf{W}[\mathbf{Q}]$ in the large $n, p, N \to \infty$ limit and (ii) numerically more accessible.

In the following theorem (proved in Appendix B), we introduce an asymptotic equivalent for $\mathbb{E}_\mathbf{W}[\mathbf{Q}]$. Instead of being directly related to the Gaussian kernel $\mathbf{K}_\mathbf{X} = \mathbf{K}_{\cos} + \mathbf{K}_{\sin}$ as suggested by (2) in the large-$N$-only limit, it depends on the two components $\mathbf{K}_{\cos}, \mathbf{K}_{\sin}$ in a more involved manner.

**Theorem 1** (Asymptotic equivalent for $\mathbb{E}_\mathbf{W}[\mathbf{Q}]$). *Under Assumption 1, for $\mathbf{Q}$ defined in (5) and $\lambda > 0$, we have, as $n \to \infty$*

$$\|\mathbb{E}_\mathbf{W}[\mathbf{Q}] - \bar{\mathbf{Q}}\| \to 0$$

*for* $\bar{\mathbf{Q}} \equiv \left( \frac{N}{n}\left( \frac{\mathbf{K}_{\cos}}{1+\delta_{\cos}} + \frac{\mathbf{K}_{\sin}}{1+\delta_{\sin}} \right) + \lambda\mathbf{I}_n \right)^{-1}$, $\mathbf{K}_{\cos} \equiv \mathbf{K}_{\cos}(\mathbf{X},\mathbf{X}), \mathbf{K}_{\sin} \equiv \mathbf{K}_{\sin}(\mathbf{X},\mathbf{X}) \in \mathbb{R}^{n \times n}$ *and*

$$\mathbf{K}_{\cos}(\mathbf{X},\mathbf{X}')_{ij} = e^{-\frac{\|\mathbf{x}_i\|^2 + \|\mathbf{x}'_j\|^2}{2}} \cosh(\mathbf{x}_i^\mathsf{T}\mathbf{x}'_j), \ \ \mathbf{K}_{\sin}(\mathbf{X},\mathbf{X}')_{ij} = e^{-\frac{\|\mathbf{x}_i\|^2 + \|\mathbf{x}'_j\|^2}{2}} \sinh(\mathbf{x}_i^\mathsf{T}\mathbf{x}'_j), \quad (6)$$

*where* $(\delta_{\cos}, \delta_{\sin})$ *is the unique positive solution to*

$$\delta_{\cos} = \frac{1}{n}\operatorname{tr}(\mathbf{K}_{\cos}\bar{\mathbf{Q}}), \quad \delta_{\sin} = \frac{1}{n}\operatorname{tr}(\mathbf{K}_{\sin}\bar{\mathbf{Q}}). \tag{7}$$

**Remark 1** (Correction to large-$N$ behavior). Taking $N/n \to \infty$, one has $\delta_{\cos} \to 0, \delta_{\sin} \to 0$ so that $\frac{\mathbf{K}_{\cos}}{1+\delta_{\cos}} + \frac{\mathbf{K}_{\sin}}{1+\delta_{\sin}} \to \mathbf{K}_{\cos} + \mathbf{K}_{\sin} = \mathbf{K}_\mathbf{X}$ and $\bar{\mathbf{Q}} \simeq \frac{n}{N}\mathbf{K}_\mathbf{X}^{-1}$, for $\lambda > 0$, in accordance with the classical large-$N$-only prediction. In this sense, the pair $(\delta_{\cos}, \delta_{\sin})$ introduced in Theorem 1 accounts for the "correction" due to the non-trivial $n/N$, as opposed to the $N \to \infty$ alone analysis. Also, when the number of features $N$ is large (i.e., as $N/n \to \infty$), the regularization effect of $\lambda$ flattens out and $\bar{\mathbf{Q}}$ behaves like (a scaled version of) the inverse Gaussian kernel matrix $\mathbf{K}_\mathbf{X}^{-1}$ (that is well-defined if $\mathbf{x}_1 \ldots, \mathbf{x}_n$ are all distinct, see [46, Theorem 2.18]).

**Remark 2** (Geometric interpretation). Since $\bar{\mathbf{Q}}$ shares the same eigenspace with $\frac{\mathbf{K}_{\cos}}{1+\delta_{\cos}} + \frac{\mathbf{K}_{\sin}}{1+\delta_{\sin}}$, one can geometrically interpret $(\delta_{\cos}, \delta_{\sin})$ as a sort of "angle" between the eigenspaces of $\mathbf{K}_{\cos}, \mathbf{K}_{\sin}$ and that of $\frac{\mathbf{K}_{\cos}}{1+\delta_{\cos}} + \frac{\mathbf{K}_{\sin}}{1+\delta_{\sin}}$. For fixed $n$, as $N \to \infty$, one has $\frac{1}{N}\sum_{t=1}^N \cos(\mathbf{X}^\mathsf{T}\mathbf{w}_t)\cos(\mathbf{w}_t^\mathsf{T}\mathbf{X}) \to \mathbf{K}_{\cos}$, $\frac{1}{N}\sum_{t=1}^N \sin(\mathbf{X}^\mathsf{T}\mathbf{w}_t)\sin(\mathbf{w}_t^\mathsf{T}\mathbf{X}) \to \mathbf{K}_{\sin}$, the eigenspaces of which are "orthogonal" to each other, so that $\delta_{\cos}, \delta_{\sin} \to 0$. On the other hand, as $N, n \to \infty$, the eigenspaces of $\mathbf{K}_{\cos}$ and $\mathbf{K}_{\sin}$ "intersect" with each other, captured by the non-trivial $(\delta_{\cos}, \delta_{\sin})$.

## 2.2 Asymptotic Training Performance

Theorem 1 provides an asymptotically more tractable approximation of $\mathbb{E}_\mathbf{W}[\mathbf{Q}]$. Together with some additional concentration arguments (e.g., from [30, Theorem 2]), this permits us to provide a complete description of the limiting behavior of the *random* bilinear form $\mathbf{a}^\mathsf{T}\mathbf{Q}\mathbf{b}$, for $\mathbf{a}, \mathbf{b} \in \mathbb{R}^n$ of bounded Euclidean norms, in such a way that $\mathbf{a}^\mathsf{T}\mathbf{Q}\mathbf{b} - \mathbf{a}^\mathsf{T}\bar{\mathbf{Q}}\mathbf{b} \xrightarrow{a.s.} 0$, as $n, p, N \to \infty$. This, together with the fact that $E_{\text{train}} = \frac{\lambda^2}{n}\mathbf{y}^\mathsf{T}\mathbf{Q}(\lambda)^2\mathbf{y} = -\frac{\lambda^2}{n}\mathbf{y}^\mathsf{T}\partial\mathbf{Q}(\lambda)\mathbf{y}/\partial\lambda$, leads to the following result on the asymptotic training error, the proof of which is given in Appendix C.

**Theorem 2** (Asymptotic training performance). *Under Assumption 1, for a given training set* $(\mathbf{X}, \mathbf{y})$ *and training MSE,* $E_{\text{train}}$ *defined in (4), as* $n \to \infty$

$$E_{\text{train}} - \bar{E}_{\text{train}} \xrightarrow{a.s.} 0, \ \bar{E}_{\text{train}} = \frac{\lambda^2}{n}\|\bar{\mathbf{Q}}\mathbf{y}\|^2 + \frac{N}{n}\frac{\lambda^2}{n^2}\left[ \frac{\operatorname{tr}(\bar{\mathbf{Q}}\mathbf{K}_{\cos}\bar{\mathbf{Q}})}{(1+\delta_{\cos})^2} \quad \frac{\operatorname{tr}(\bar{\mathbf{Q}}\mathbf{K}_{\sin}\bar{\mathbf{Q}})}{(1+\delta_{\sin})^2} \right] \mathbf{\Omega} \begin{bmatrix} \mathbf{y}^\mathsf{T}\bar{\mathbf{Q}}\mathbf{K}_{\cos}\bar{\mathbf{Q}}\mathbf{y} \\ \mathbf{y}^\mathsf{T}\bar{\mathbf{Q}}\mathbf{K}_{\sin}\bar{\mathbf{Q}}\mathbf{y} \end{bmatrix}$$

*for* $\bar{\mathbf{Q}}$ *defined in Theorem 1 and*

$$\mathbf{\Omega}^{-1} \equiv \mathbf{I}_2 - \frac{N}{n}\begin{bmatrix} \frac{1}{n}\frac{\operatorname{tr}(\bar{\mathbf{Q}}\mathbf{K}_{\cos}\bar{\mathbf{Q}}\mathbf{K}_{\cos})}{(1+\delta_{\cos})^2} & \frac{1}{n}\frac{\operatorname{tr}(\bar{\mathbf{Q}}\mathbf{K}_{\cos}\bar{\mathbf{Q}}\mathbf{K}_{\sin})}{(1+\delta_{\sin})^2} \\ \frac{1}{n}\frac{\operatorname{tr}(\bar{\mathbf{Q}}\mathbf{K}_{\cos}\bar{\mathbf{Q}}\mathbf{K}_{\sin})}{(1+\delta_{\cos})^2} & \frac{1}{n}\frac{\operatorname{tr}(\bar{\mathbf{Q}}\mathbf{K}_{\sin}\bar{\mathbf{Q}}\mathbf{K}_{\sin})}{(1+\delta_{\sin})^2} \end{bmatrix}. \tag{8}$$

One can show that (i) for a given $n$ and $\lambda > 0$, $\bar{E}_{\text{train}}$ decreases as the model size $N$ increases; and (ii) for a given ratio $N/n$, $\bar{E}_{\text{train}}$ increases as the regularization penalty $\lambda$ grows large, as expected.

## 2.3 Asymptotic Test Performance

Theorem 2 holds without any restriction on the training set, $(\mathbf{X}, \mathbf{y})$, except for Assumption 1, since only the randomness of $\mathbf{W}$ is involved, and thus one can simply treat $(\mathbf{X}, \mathbf{y})$ as known in this result. This is no longer the case for the test error. Intuitively, the test data $\hat{\mathbf{X}}$ cannot be chosen arbitrarily, and one must ensure that the test data "behave" statistically like the training data, in a "well-controlled" manner, so that the test MSE is asymptotically deterministic and bounded as $n, \hat{n}, p, N \to \infty$. Following this intuition, we work under the following assumption.

**Assumption 2** (Data as concentrated random vectors [29]). *The training data* $\mathbf{x}_i \in \mathbb{R}^p, i \in \{1, \ldots, n\}$, *are independently drawn (non-necessarily uniformly) from one of* $K > 0$ *distribution*

classes[2] $\mu_1, \ldots, \mu_K$. *There exist constants $C, \eta, q > 0$ such that for any $\mathbf{x}_i \sim \mu_k, k \in \{1, \ldots, K\}$ and any 1-Lipschitz function $f : \mathbb{R}^p \to \mathbb{R}$, we have*

$$\mathbb{P}\left(\|f(\mathbf{x}_i) - \mathbb{E}[f(\mathbf{x}_i)]\| \geq t\right) \leq Ce^{-(t/\eta)^q}, \quad t \geq 0. \tag{9}$$

*The test data $\hat{\mathbf{x}}_i \sim \mu_k$, $i \in \{1, \ldots, \hat{n}\}$ are mutually independent, but* may depend on *training data $\mathbf{X}$ and $\|\mathbb{E}[\sigma(\mathbf{WX}) - \sigma(\mathbf{W\hat{X}})]\| = O(\sqrt{n})$ for $\sigma \in \{\cos, \sin\}$.*

To facilitate the discussion of the phase transition and the double descent, we do not assume independence between training data and test data (but we do assume independence between different columns within $\mathbf{X}$ and $\hat{\mathbf{X}}$). In this respect, Assumption 2 is weaker than the classical i.i.d. assumption, and it permits us to illustrate the impact of training-test similarity on the model performance (Section 3.3).

A first example of concentrated random vectors satisfying (9) is the random Gaussian vector $\mathcal{N}(\mathbf{0}, \mathbf{I}_p)$ [24]. Moreover, since the concentration property in (9) is stable over Lipschitz transformations [29], it holds, for any 1-Lipschitz mapping $g : \mathbb{R}^d \to \mathbb{R}^p$ and $\mathbf{z} \sim \mathcal{N}(\mathbf{0}, \mathbf{I}_d)$, that $g(\mathbf{z})$ also satisfies (9). In this respect, Assumption 2, although seemingly quite restrictive, represents a large family of "generative models", including notably the "fake images" generated by modern generative adversarial networks (GANs) that are, by construction, Lipschitz transformations of large random Gaussian vectors [18, 47]. As such, from a practical consideration, Assumption 2 provides a more realistic and flexible statistical model for real-world data.

With Assumption 2, we have the following result on the asymptotic test error, proved in Section D.

**Theorem 3** (Asymptotic test performance). *Under Assumptions 1 and 2, we have, for test MSE $E_{\text{test}}$ defined in (4) and test data $(\hat{\mathbf{X}}, \hat{\mathbf{y}})$ satisfying $\limsup_{\hat{n}} \|\hat{\mathbf{X}}\| < \infty$, $\limsup_{\hat{n}} \|\hat{\mathbf{y}}\|_\infty < \infty$ with $\hat{n}/n \in (0, \infty)$ that, as $n \to \infty$*

$$E_{\text{test}} - \bar{E}_{\text{test}} \xrightarrow{a.s.} 0, \; \bar{E}_{\text{test}} = \frac{1}{\hat{n}}\|\hat{\mathbf{y}} - \frac{N}{n}\hat{\boldsymbol{\Phi}}\bar{\mathbf{Q}}\mathbf{y}\|^2 + \frac{N^2}{n^2\hat{n}} \begin{bmatrix} \frac{\Theta_{\cos}}{(1+\delta_{\cos})^2} & \frac{\Theta_{\sin}}{(1+\delta_{\sin})^2} \end{bmatrix} \boldsymbol{\Omega} \begin{bmatrix} \mathbf{y}^\mathsf{T}\bar{\mathbf{Q}}\mathbf{K}_{\cos}\bar{\mathbf{Q}}\mathbf{y} \\ \mathbf{y}^\mathsf{T}\bar{\mathbf{Q}}\mathbf{K}_{\sin}\bar{\mathbf{Q}}\mathbf{y} \end{bmatrix}$$

*for $\boldsymbol{\Omega}$ defined in (8),*

$$\Theta_\sigma = \frac{1}{N}\operatorname{tr}\mathbf{K}_\sigma(\hat{\mathbf{X}}, \hat{\mathbf{X}}) + \frac{N}{n}\frac{1}{n}\operatorname{tr}\bar{\mathbf{Q}}\hat{\boldsymbol{\Phi}}^\mathsf{T}\hat{\boldsymbol{\Phi}}\bar{\mathbf{Q}}\mathbf{K}_\sigma - \frac{2}{n}\operatorname{tr}\bar{\mathbf{Q}}\hat{\boldsymbol{\Phi}}^\mathsf{T}\mathbf{K}_\sigma(\hat{\mathbf{X}}, \mathbf{X}), \quad \sigma \in \{\cos, \sin\}, \tag{10}$$

*and $\boldsymbol{\Phi} \equiv \frac{\mathbf{K}_{\cos}}{1+\delta_{\cos}} + \frac{\mathbf{K}_{\sin}}{1+\delta_{\sin}}$, $\hat{\boldsymbol{\Phi}} \equiv \frac{\mathbf{K}_{\cos}(\hat{\mathbf{X}}, \mathbf{X})}{1+\delta_{\cos}} + \frac{\mathbf{K}_{\sin}(\hat{\mathbf{X}}, \mathbf{X})}{1+\delta_{\sin}}$, with $\mathbf{K}_{\cos}(\hat{\mathbf{X}}, \mathbf{X}), \mathbf{K}_{\sin}(\hat{\mathbf{X}}, \mathbf{X}) \in \mathbb{R}^{\hat{n} \times n}$ and $\mathbf{K}_{\cos}(\hat{\mathbf{X}}, \hat{\mathbf{X}}), \mathbf{K}_{\sin}(\hat{\mathbf{X}}, \hat{\mathbf{X}}) \in \mathbb{R}^{\hat{n} \times \hat{n}}$ defined as in (6).*

Taking $(\hat{\mathbf{X}}, \hat{\mathbf{y}}) = (\mathbf{X}, \mathbf{y})$, one gets $\bar{E}_{\text{test}} = \bar{E}_{\text{train}}$, as expected. From this perspective, Theorem 3 can be seen as an extension of Theorem 2, with the "interaction" between training and test data (i.e., training-versus-test $\mathbf{K}_\sigma(\hat{\mathbf{X}}, \mathbf{X})$ and test-versus-test $\mathbf{K}_\sigma(\hat{\mathbf{X}}, \hat{\mathbf{X}})$ interaction matrices) summarized in the scalar parameter $\Theta_\sigma$ defined in (10), for $\sigma \in \{\cos, \sin\}$.

# 3 Empirical Evaluations and Practical Implications

In this section, we provide a detailed empirical evaluation, including a discussion of the behavior of the fixed-point equation in Theorem 1, and its consequences in Theorem 2 and Theorem 3. In particular, we describe the behavior of the pair $(\delta_{\cos}, \delta_{\sin})$ that characterizes the necessary correction in the large $n, p, N$ regime, as a function of the regularization $\lambda$ and the ratio $N/n$. This explains: (i) the mismatch between empirical regression errors from the Gaussian kernel prediction (Figure 1); and (ii) the behavior of $(\delta_{\cos}, \delta_{\sin})$ as a function of $N/n$, which clearly indicates two phases of learning (Figure 3) and the corresponding double descent test error curves (Figure 4).

## 3.1 Correction due to the Large $n, p, N$ Regime

The RFF Gram matrix $\boldsymbol{\Sigma}_\mathbf{X}^\mathsf{T}\boldsymbol{\Sigma}_\mathbf{X}/N$ is *not* close to the classical Gaussian kernel matrix $\mathbf{K_X}$ in the large $n, p, N$ regime; and, as a consequence, its resolvent $\mathbf{Q}$, as well the training and test MSE, $E_{\text{train}}$ and $E_{\text{test}}$ (that are functions of $\mathbf{Q}$), behave quite differently from the Gaussian kernel predictions.

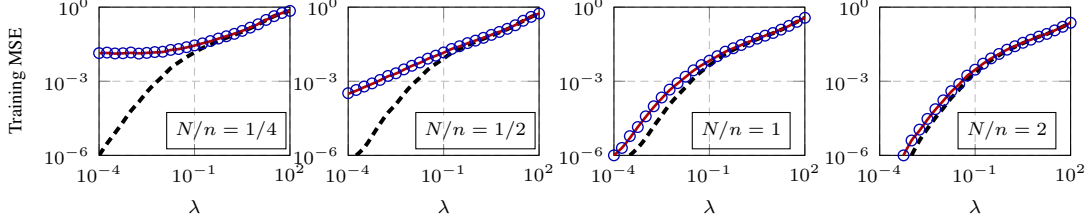

Figure 1: Training MSEs of RFF ridge regression on MNIST data (class 3 versus 7), as a function of regression penalty $\lambda$, for $p = 784$, $n = 1\,000$, $N = 250, 500, 1\,000, 2\,000$. Empirical results displayed in **blue** circles; Gaussian kernel predictions (assuming $N \to \infty$ alone) in **black** dashed lines; and Theorems 2 in **red** solid lines. Results obtained by averaging over 30 runs.

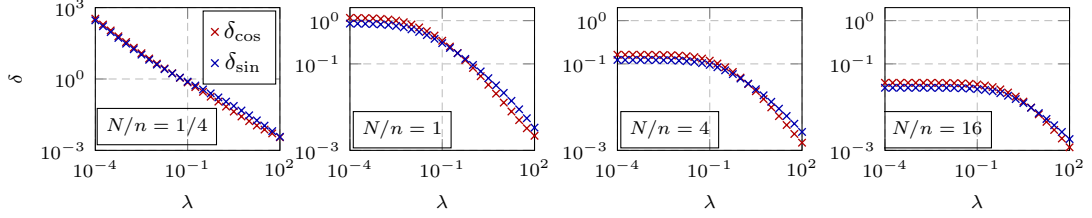

Figure 2: Behavior of $(\delta_{\cos}, \delta_{\sin})$ in (11) on MNIST data (class 3 versus 7), as a function of the regularization parameter $\lambda$, for $p = 784$, $n = 1\,000$, $N = 250, 1\,000, 4\,000, 16\,000$.

As already discussed in Remark 1 after Theorem 1, for $\lambda > 0$, the pair $(\delta_{\cos}, \delta_{\sin})$ characterizes the correction when considering $n, p, N$ all large, compared to the large-$N$-only asymptotic behavior:

$$\delta_{\cos} = \frac{1}{n} \operatorname{tr} \mathbf{K}_{\cos} \bar{\mathbf{Q}}, \ \delta_{\sin} = \frac{1}{n} \operatorname{tr} \mathbf{K}_{\sin} \bar{\mathbf{Q}}, \quad \bar{\mathbf{Q}} = \left( \frac{N}{n} \left( \frac{\mathbf{K}_{\cos}}{1 + \delta_{\cos}} + \frac{\mathbf{K}_{\sin}}{1 + \delta_{\sin}} \right) + \lambda \mathbf{I}_n \right)^{-1}. \quad (11)$$

To start, Figure 1 compares the training MSEs of RFF ridge regression to the predictions from Gaussian kernel regression and to the predictions from our Theorem 2, on the popular MNIST data set [23]. Observe that there is a huge gap between empirical training errors and the Gaussian kernel predictions, especially when $N/n < 1$, while our theory *consistently* fits empirical observations almost perfectly.

Next, from (11) we know that both $\delta_{\cos}$ and $\delta_{\sin}$ are decreasing functions of $\lambda$. (See Lemma 7 in Appendix E for a proof of this fact.) Figure 2 shows that: (i) over a range of different $N/n$, both $\delta_{\cos}$ and $\delta_{\sin}$ decrease monotonically as $\lambda$ increases; (ii) the behavior for $N/n < 1$, which is decreasing from an initial value of $\delta \gg 1$, is very different from the behavior for $N/n \gtrsim 1$, where an initially flat region is observed for small values of $\lambda$ and we have $\delta < 1$ for all values of $\lambda$; and (iii) the impact of regularization $\lambda$ becomes less significant as the ratio $N/n$ becomes large. This is in accordance with the limiting behavior of $\bar{\mathbf{Q}} \simeq \frac{n}{N} \mathbf{K}_{\mathbf{X}}^{-1}$ in Remark 1 that is *independent* of $\lambda$ as $N/n \to \infty$.

Note also that, while $\delta_{\cos}$ and $\delta_{\sin}$ can be geometrically interpreted as a sort of weighted "angle" between different kernel matrices (as in Remark 2), and therefore one might expect to have $\delta \in [0, 1]$, this is not the case for the leftmost plot of Figure 1 with $N/n = 1/4$. There, for small values of $\lambda$ (say $\lambda \lesssim 0.1$), both $\delta_{\cos}$ and $\delta_{\sin}$ scale like $\lambda^{-1}$, while they are observed to saturate to a fixed $O(1)$ value for $N/n = 1, 4, 16$. This corresponds to two different phases of learning in the ridgeless $\lambda \to 0$ limit, as discussed in the following section.

## 3.2 Phase Transition and Corresponding Double Descent

Both $\delta_{\cos}$ and $\delta_{\sin}$ in (11) are decreasing functions of $N$, as depicted in Figure 3. (See Lemma 6 in Appendix E for a proof.) More importantly, Figure 3 also illustrates that $\delta_{\cos}$ and $\delta_{\sin}$ exhibit qualitatively different behavior, depending on the ratio $N/n$. For $\lambda$ not too small ($\lambda = 1$ or $10$), both $\delta_{\cos}$ and $\delta_{\sin}$ decrease *smoothly*, as $N/n$ grows large. However, for $\lambda$ relatively small ($\lambda = 10^{-3}$ and $10^{-7}$), we observe a "phase transition" on two sides of the interpolation threshold $2N = n$. (Note

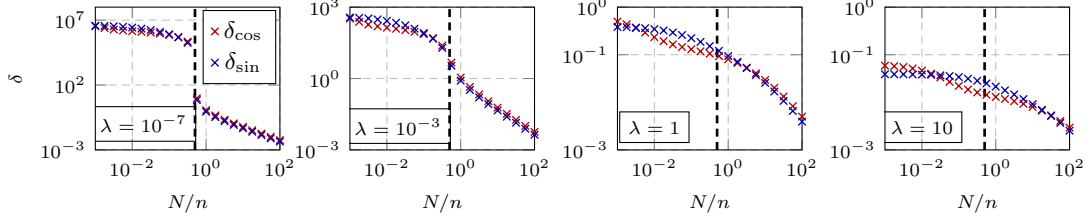

Figure 3: Behavior of $(\delta_{\cos}, \delta_{\sin})$ on MNIST data (class 3 versus 7), as a function of $N/n$, $p = 784$, $n = 1\,000$, $\lambda = 10^{-7}, 10^{-3}, 1, 10$. The **black** dashed line is the interpolation threshold $2N = n$.

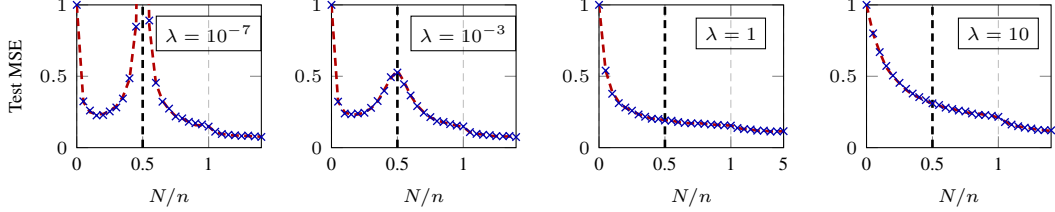

Figure 4: Empirical (**blue** crosses) and theoretical (**red** dashed lines) test errors of RFF regression as a function of the ratio $N/n$, on MNIST data set (class 3 versus 7), for $p = 784$, $n = 500$, $\lambda = 10^{-7}, 10^{-3}, 1, 10$. The **black** dashed line is the interpolation threshold $2N = n$.

that the scale of the y-axis is different in different subfigures.) More precisely, in the leftmost plot with $\lambda = 10^{-7}$, $\delta_{\cos}$ and $\delta_{\sin}$ "jump" from order $O(1)$ (when $2N > n$) to much higher values of the same order of $\lambda^{-1}$ (when $2N < n$). A similar behavior is also observed for $\lambda = 10^{-3}$.

This phase transition can be theoretically justified by considering the *ridgeless* $\lambda \to 0$ limit in Theorem 1. First note that, for $\lambda = 0$ and $2N < n$, the (random) resolvent $\mathbf{Q}(\lambda = 0)$ in (5) is simply undefined, as it involves inverting a singular matrix $\mathbf{\Sigma_X^\top \Sigma_X} \in \mathbb{R}^{n \times n}$ that is of rank at most $2N < n$. As a consequence, we expect to see both $\mathbf{Q}$ and $\bar{\mathbf{Q}}$ scale like $\lambda^{-1}$ as $\lambda \to 0$ for $2N < n$, while for $2N > n$ this is no longer the case. As a consequence, we have the following two phases:

1.*Under-parameterized* with $2N < n$. Here, $\mathbf{Q}$ is not well-defined (indeed $\mathbf{Q}$ scales like $\lambda^{-1}$) and one must consider instead the properly scaled $\lambda\delta_{\cos}, \lambda\delta_{\sin}$ and $\lambda\bar{\mathbf{Q}}$ as $\lambda \to 0$.

2.*Over-parameterized* with $2N > n$, where one can take $\lambda \to 0$ in (11) to get $\delta_{\cos}, \delta_{\sin}$ and $\bar{\mathbf{Q}}$.

**Remark 3** (Double descent test error curves). On account of the above two phases, it is not surprising to observe a "singular" behavior at $2N = n$, when no regularization is applied. Here, we consider the (asymptotic) test MSE in Theorem 3 in the ridgeless $\lambda \to 0$ limit and focus on the situation where the test data $\hat{\mathbf{X}}$ is sufficiently different from the training data $\mathbf{X}$ (see more discussions on this point in Section 3.3 below). Then, the two-by-two matrix $\mathbf{\Omega}$ defined in (8) diverges to infinity at $2N = n$ as $\lambda \to 0$. (Indeed, the determinant $\det(\mathbf{\Omega}^{-1})$ scales as $\lambda$, per Lemma 5 in Appendix E.) As a consequence, we have $\bar{E}_{\text{test}} \to \infty$ as $N/n \to 1/2$, resulting in a sharp deterioration in the test performance around the interpolation threshold $2N = n$. It is also interesting to note that, while $\mathbf{\Omega}$ also appears in $\bar{E}_{\text{train}}$, we still obtain (asymptotically) zero training MSE at $2N = n$, despite the divergence of $\mathbf{\Omega}$ as $\lambda \to 0$, essentially due to the prefactor $\lambda^2$ in $\bar{E}_{\text{train}}$.

Figure 4 depicts the empirical and theoretical test MSEs with different $\lambda$. In particular, for $\lambda = 10^{-7}$ and $\lambda = 10^{-3}$, a double-descent-type behavior is observed, with a singularity at $2N = n$, while for larger values of $\lambda$ ($\lambda = 1$ and $10$), a smoother and monotonically decreasing test error curve is observed, as a function of $N/n$, in accordance with the observations in [36] on Gaussian data.

**Remark 4** (Double descent as a consequence of phase transition). While the double descent phenomenon has received considerable attention recently, our analysis makes it clear that in this model (and presumably many others) it is a natural consequence of the phase transition between two qualitatively different phases of learning [32].

### 3.3 Impact of Training-test Similarity

We see that the (asymptotic) test error behaves entirely differently, depending on whether the test data $\hat{\mathbf{X}}$ is "close to" the training data $\mathbf{X}$ or not. For $\hat{\mathbf{X}} = \mathbf{X}$, one has $\bar{E}_{\text{test}} = \bar{E}_{\text{train}}$ that decreases monotonically as $N$ grows large; while for $\hat{\mathbf{X}}$ sufficiently different from $\mathbf{X}$ (in the associated kernel space in the sense that $\mathbf{K}_\sigma(\mathbf{X}, \mathbf{X})$ is sufficiently different from $\mathbf{K}_\sigma(\hat{\mathbf{X}}, \mathbf{X})$ for $\sigma \in \{\cos, \sin\}$), $\bar{E}_{\text{test}}$ diverges at $2N = n$ and establishes a double descent behavior. To have a more quantitative assessment of the impact of training-test similarity on the RFF model performance, we consider here the special case $\hat{\mathbf{y}} = \mathbf{y}$. Since in the ridgeless $\lambda \to 0$ limit, $\mathbf{\Omega}$ scales as $\lambda^{-1}$ at $2N = n$ (Remark 3), one must then have $\Theta_\sigma \propto \lambda$ to "compensate" so that $\bar{E}_{\text{test}}$ does not diverge at $2N = n$ as $\lambda \to 0$. A first example is the case where the test data is a small perturbation of the training data. In Figure 5, the test data are generated by adding Gaussian white noise of variance $\sigma^2$ to the training data, i.e.,

$$\hat{\mathbf{x}}_i = \mathbf{x}_i + \sigma \varepsilon_i \tag{12}$$

for independent $\varepsilon_i \sim \mathcal{N}(\mathbf{0}, \mathbf{I}_p/p)$. In Figure 5, we observe that (i) below the threshold $\sigma^2 = \lambda$, the test error coincides with the training error and both are relatively small for $2N = n$; and (ii) as soon as $\sigma^2 > \lambda$, the test error diverges from the training error and grows large (but linearly in $\sigma^2$) as the noise level increases. Note also from the two rightmost plots of Figure 5 that the training-to-test "transition" at $\sigma^2 \simeq \lambda$ is *sharp* only for relatively small values of $\lambda$, as predicted by our theory.

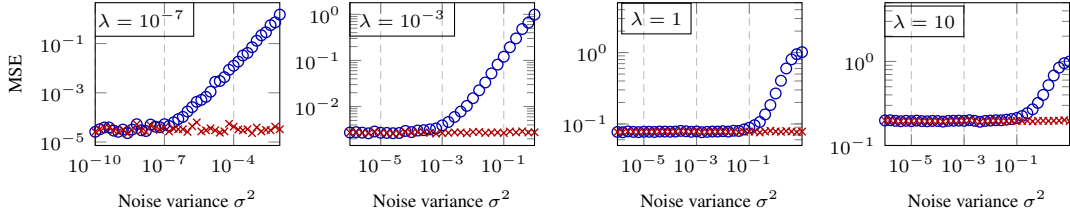

Figure 5: Empirical training (**red** crosses) and test (**blue** circles) errors of RFF ridge regression on MNIST data (class 3 versus 7), as a function of the noise level $\sigma^2$, for $N = 512$, $p = 784$, $n = \hat{n} = 1\,024 = 2N$, $\lambda = 10^{-7}, 10^{-3}, 1, 10$. Results obtained by averaging over 30 runs.

## 4   Conclusion

We have established a precise description of the resolvent of RFF Gram matrices, and provided asymptotic training and test performance guarantees for RFF ridge regression, in the $n, p, N \to \infty$ limit. We have also discussed the under- and over-parameterized regimes, where the resolvent behaves dramatically differently. These observations involve only mild regularity assumptions on the data, yielding phase transition behavior and double descent test error curves for RFF regression that closely match experiments on real-world data. Extended to a (technically more involved) multi-layer setting in the more realistic large $n, p, N$ regime as in [16], our analysis may shed new light on the theoretical understanding of modern deep neural nets, beyond the large-$N$ alone neural tangent kernel limit.

## Broader Impact

In this article, we provide theoretical assessment of the popular random Fourier features (RFFs), in the practical setting where $n, p, N$ are all large and comparable. Asymptotic performance guarantees are provided for RFF ridge regression in this $n, p, N \to \infty$ limit, as an important positive impact of this work on the development of more reliable large-scale machine learning systems. The theoretical framework developed in this article presents fair and non-offensive societal consequence.

**Acknowledgments.**   We would like to acknowledge the UC Berkeley CLTC, ARO, IARPA, NSF, and ONR for providing partial support of this work. Our conclusions do not necessarily reflect the position or the policy of our sponsors, and no official endorsement should be inferred. Couillet's work is partially supported by MIAI at University Grenoble-Alpes (ANR-19-P3IA-0003).

## Footnotes

[1]For readers not familiar with the impact of spectral norm error in learning, or with the random matrix theory techniques that we will use in our analysis, such as resolvent analysis and the use of deterministic equivalents, see Appendix A for a warm-up discussion.

[2]$K \geq 2$ is included to cover multi-class classification problems; and $K$ should remain fixed as $n, p \to \infty$.

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
