[Supplementary Material]

# A   Warm-up Example: Sample Covariance and Marčenko-Pastur Equation

Consider the sample covariance matrix $\hat{\mathbf{C}} = \frac{1}{n}\mathbf{X}\mathbf{X}^{\mathsf{T}}$ from some data $\mathbf{X} \in \mathbb{R}^{p \times n}$ composed of $n$ i.i.d. $\mathbf{x}_i \sim \mathcal{N}(\mathbf{0}, \mathbf{C})$ with positive definite $\mathbf{C} \in \mathbb{R}^{p \times p}$. In this zero-mean Gaussian setting, the sample covariance $\hat{\mathbf{C}}$, despite being the maximum likelihood estimator of the *population covariance* $\mathbf{C}$ and providing *entry-wise* consistent estimate for it, is an extremely poor estimator of $\mathbf{C}$ in a *spectral norm* sense, for $n, p$ large. More precisely, $\|\hat{\mathbf{C}} - \mathbf{C}\| \not\to 0$ as $n, p \to \infty$ with $p/n \to c \in (0, \infty)$. Indeed, one has $\|\hat{\mathbf{C}} - \mathbf{C}\|/\|\mathbf{C}\| \approx 20\%$, even with $n = 100p$, in the simple $\mathbf{C} = \mathbf{I}_p$ setting. Figure 6 compares the eigenvalue histogram of $\hat{\mathbf{C}}$ with the population eigenvalue of $\mathbf{C}$, in the setting of $\mathbf{C} = \mathbf{I}_p$ and $n = 100p$. In the $\mathbf{C} = \mathbf{I}_p$ case, the limiting eigenvalue distribution of $\hat{\mathbf{C}}$ as $n, p \to \infty$ is known to be the popular Marčenko-Pastur law [31] given by

$$\mu(dx) = (1 - c^{-1}) \cdot \delta_0(x) + \frac{1}{2\pi c x}\sqrt{\left(x - (1 - \sqrt{c})^2\right)^+ \left((1 + \sqrt{c})^2 - x\right)^+}\,dx \qquad (13)$$

with $\delta_0(x)$ the Dirac mass at zero, $c = \lim p/n$ and $(x)^+ = \max(x, 0)$, so that the support of $\mu$ has length $(1 + \sqrt{c})^2 - (1 - \sqrt{c})^2 = 4\sqrt{c} = 0.4$ for $n = 100p$.

Figure 6: Eigenvalue histogram of $\hat{\mathbf{C}}$ versus the Marčenko-Pastur law, for $p = 512$ and $n = 100p$.

In the regression analysis (such as ridge regression) based on $\mathbf{X}$, of more immediate interest is the *resolvent* $\mathbf{Q}_{\hat{\mathbf{C}}}(\lambda) \equiv (\hat{\mathbf{C}} + \lambda\mathbf{I}_p)^{-1}, \lambda > 0$ of the sample covariance $\hat{\mathbf{C}}$, and more concretely, the bilinear forms of the type $\mathbf{a}^{\mathsf{T}}\mathbf{Q}_{\hat{\mathbf{C}}}(\lambda)\mathbf{b}$ for $\mathbf{a}, \mathbf{b} \in \mathbb{R}^p$. As a result of the spectral norm inconsistency $\|\hat{\mathbf{C}} - \mathbf{C}\| \not\to 0$ in the large $n, p$ regime, it is unlikely that for most $\mathbf{a}, \mathbf{b}$, the convergence $\mathbf{a}^{\mathsf{T}}\mathbf{Q}_{\hat{\mathbf{C}}}(\lambda)\mathbf{b} - \mathbf{a}^{\mathsf{T}}(\mathbf{C} + \lambda\mathbf{I}_p)^{-1}\mathbf{b} \to 0$ would still hold.

While the *random* variable $\mathbf{a}^{\mathsf{T}}\mathbf{Q}_{\hat{\mathbf{C}}}(\lambda)\mathbf{b}$ is not getting close to $\mathbf{a}^{\mathsf{T}}(\mathbf{C} + \lambda\mathbf{I}_p)^{-1}\mathbf{b}$ as $n, p \to \infty$, it does exhibit a tractable asymptotically *deterministic* behavior, described by the Marčenko-Pastur equation [31] for $\mathbf{C} = \mathbf{I}_p$. Notably, for $\mathbf{a}, \mathbf{b} \in \mathbb{R}^p$ deterministic vectors of bounded Euclidean norms, we have, as $n, p \to \infty$ and $p/n \to c \in (0, \infty)$,

$$\mathbf{a}^{\mathsf{T}}\mathbf{Q}_{\hat{\mathbf{C}}}(\lambda)\mathbf{b} - m(\lambda) \cdot \mathbf{a}^{\mathsf{T}}\mathbf{b} \xrightarrow{a.s.} 0,$$

with $m(\lambda)$ the unique positive solution to the following Marčenko-Pastur equation [31]

$$c\lambda m^2(\lambda) + (1 + \lambda - c)m(\lambda) - 1 = 0. \qquad (14)$$

In a sense, $\bar{\mathbf{Q}}(\lambda) \equiv m(\lambda)\mathbf{I}_p$ can be seen as a *deterministic equivalent* [19, 11] for the *random* $\mathbf{Q}_{\hat{\mathbf{C}}}(\lambda)$ that asymptotically characterizes the behavior of the latter, when bilinear forms are considered. In Figure 7 we compare the quadratic forms $\mathbf{a}^{\mathsf{T}}\mathbf{Q}_{\hat{\mathbf{C}}}(\lambda)\mathbf{a}$ as a function of $\lambda$, for $n = 10p$ and $n = 2p$. We observe that, in both cases, the RMT prediction in (14) provides a much closer match than the large-$n$ alone asymptotic given by $\mathbf{a}^{\mathsf{T}}(\mathbf{C} + \lambda\mathbf{I}_p)^{-1}\mathbf{a}$. This, together with Figure 1 on RFF ridge regression model, conveys a strong practical motivation of this work.

Figure 7: Quadratic forms $\mathbf{a}^\mathsf{T}\mathbf{Q}_{\hat{\mathbf{C}}}(\lambda)\mathbf{a}$ as a function of $\lambda$, for $p = 512$, $n = 10p$ (**left**) and $n = 2p$ (**right**). Empirical results displayed in **blue** circles; population predictions $\mathbf{a}^\mathsf{T}(\mathbf{C} + \lambda\mathbf{I}_p)^{-1}\mathbf{a}$ (assuming $n \to \infty$ alone with $p$ fixed) in **black** dashed lines; and RMT prediction from (14) in **red** solid lines. Results obtained by averaging over 50 runs.

# B    Proof of Theorem 1

Our objective is to prove, under Assumption 1, the asymptotic equivalence between the expectation (with respect to $\mathbf{W}$, omitted from now on) $\mathbb{E}[\mathbf{Q}]$ and

$$\bar{\mathbf{Q}} \equiv \left( \frac{N}{n} \left( \frac{\mathbf{K}_{\cos}}{1 + \delta_{\cos}} + \frac{\mathbf{K}_{\sin}}{1 + \delta_{\sin}} \right) + \lambda\mathbf{I}_n \right)^{-1}$$

for $\mathbf{K}_{\cos} \equiv \mathbf{K}_{\cos}(\mathbf{X}, \mathbf{X}), \mathbf{K}_{\sin} \equiv \mathbf{K}_{\sin}(\mathbf{X}, \mathbf{X}) \in \mathbb{R}^{n \times n}$ defined in (6), with $(\delta_{\cos}, \delta_{\cos})$ the unique positive solution to

$$\delta_{\cos} = \frac{1}{n}\operatorname{tr}(\mathbf{K}_{\cos}\bar{\mathbf{Q}}), \quad \delta_{\sin} = \frac{1}{n}\operatorname{tr}(\mathbf{K}_{\sin}\bar{\mathbf{Q}}).$$

The existence and uniqueness of the above fixed-point equation is standard in random matrix literature and can be reached for instance with the standard interference function framework [54].

The asymptotic equivalence should be announced in the sense that $\|\mathbb{E}[\mathbf{Q}] - \bar{\mathbf{Q}}\| \to 0$ as $n, p, N \to \infty$ at the same pace. We shall proceed by introducing an intermediary resolvent $\tilde{\mathbf{Q}}$ (see definition in (16)) and show subsequently that

$$\|\mathbb{E}[\mathbf{Q}] - \tilde{\mathbf{Q}}\| \to 0, \quad \|\tilde{\mathbf{Q}} - \bar{\mathbf{Q}}\| \to 0.$$

In the sequel, we use $o(1)$ and $o_{\|\cdot\|}(1)$ for scalars or matrices of (almost surely if being random) vanishing absolute values or operator norms as $n, p \to \infty$.

We start by introducing the following lemma.

**Lemma 1** (Expectation of $\sigma_1(\mathbf{x}_i^\mathsf{T}\mathbf{w})\sigma_2(\mathbf{w}^\mathsf{T}\mathbf{x}_j)$). *For $\mathbf{w} \sim \mathcal{N}(\mathbf{0}, \mathbf{I}_p)$ and $\mathbf{x}_i, \mathbf{x}_j \in \mathbb{R}^p$ we have (per Definition in (6))*

$$\mathbb{E}_{\mathbf{w}}[\cos(\mathbf{x}_i^\mathsf{T}\mathbf{w})\cos(\mathbf{w}^\mathsf{T}\mathbf{x}_j)] = e^{-\frac{1}{2}(\|\mathbf{x}_i\|^2 + \|\mathbf{x}_j\|^2)}\cosh(\mathbf{x}_i^\mathsf{T}\mathbf{x}_j) \equiv [\mathbf{K}_{\cos}(\mathbf{X}, \mathbf{X})]_{ij} \equiv [\mathbf{K}_{\cos}]_{ij}$$

$$\mathbb{E}_{\mathbf{w}}[\sin(\mathbf{x}_i^\mathsf{T}\mathbf{w})\sin(\mathbf{w}^\mathsf{T}\mathbf{x}_j)] = e^{-\frac{1}{2}(\|\mathbf{x}_i\|^2 + \|\mathbf{x}_j\|^2)}\sinh(\mathbf{x}_i^\mathsf{T}\mathbf{x}_j) \equiv [\mathbf{K}_{\sin}(\mathbf{X}, \mathbf{X})]_{ij} \equiv [\mathbf{K}_{\sin}]_{ij}$$

$$\mathbb{E}_{\mathbf{w}}[\cos(\mathbf{x}_i^\mathsf{T}\mathbf{w})\sin(\mathbf{w}^\mathsf{T}\mathbf{x}_j)] = 0.$$

*Proof of Lemma 1.* The proof follows the integration tricks in [52, 30]. Note in particular that the third equality holds in the case of $(\cos, \sin)$ nonlinearity but in general not true for arbitrary Lipschitz $(\sigma_1, \sigma_2)$. □

Let us focus on the resolvent $\mathbf{Q} \equiv \left( \frac{1}{n}\mathbf{\Sigma}_\mathbf{X}^\mathsf{T}\mathbf{\Sigma}_\mathbf{X} + \lambda\mathbf{I}_n \right)^{-1}$ of $\frac{1}{n}\mathbf{\Sigma}_\mathbf{X}^\mathsf{T}\mathbf{\Sigma}_\mathbf{X} \in \mathbb{R}^{n \times n}$, for random Fourier feature matrix $\mathbf{\Sigma}_\mathbf{X} \equiv \begin{bmatrix} \cos(\mathbf{W}\mathbf{X}) \\ \sin(\mathbf{W}\mathbf{X}) \end{bmatrix}$ that can be rewritten as

$$\mathbf{\Sigma}_\mathbf{X}^\mathsf{T} = [\cos(\mathbf{X}^\mathsf{T}\mathbf{w}_1), \dots, \cos(\mathbf{X}^\mathsf{T}\mathbf{w}_N), \sin(\mathbf{X}^\mathsf{T}\mathbf{w}_1), \dots, \sin(\mathbf{X}^\mathsf{T}\mathbf{w}_N)] \tag{15}$$

for $\mathbf{w}_i$ the $i$-th row of $\mathbf{W} \in \mathbb{R}^{N \times p}$ with $\mathbf{w}_i \sim \mathcal{N}(\mathbf{0}, \mathbf{I}_p), i = 1, \ldots, N$, that is at the core of our analysis. Note from (15) that we have

$$\mathbf{\Sigma_X^\mathsf{T} \Sigma_X} = \sum_{i=1}^{N} \left( \cos(\mathbf{X^\mathsf{T} w}_i) \cos(\mathbf{w}_i^\mathsf{T} \mathbf{X}) + \sin(\mathbf{X^\mathsf{T} w}_i) \sin(\mathbf{w}_i^\mathsf{T} \mathbf{X}) \right) = \sum_{i=1}^{N} \mathbf{U}_i \mathbf{U}_i^\mathsf{T}$$

with $\mathbf{U}_i = \begin{bmatrix} \cos(\mathbf{X^\mathsf{T} w}_i) & \sin(\mathbf{X^\mathsf{T} w}_i) \end{bmatrix} \in \mathbb{R}^{n \times 2}$.

Letting

$$\tilde{\mathbf{Q}} \equiv \left( \frac{N}{n} \frac{\mathbf{K}_{\cos}}{1 + \alpha_{\cos}} + \frac{N}{n} \frac{\mathbf{K}_{\sin}}{1 + \alpha_{\sin}} + \lambda \mathbf{I}_n \right)^{-1} \tag{16}$$

with

$$\alpha_{\cos} = \frac{1}{n} \operatorname{tr}(\mathbf{K}_{\cos} \mathbb{E}[\mathbf{Q}]), \quad \alpha_{\sin} = \frac{1}{n} \operatorname{tr}(\mathbf{K}_{\sin} \mathbb{E}[\mathbf{Q}]) \tag{17}$$

we have, with the resolvent identity $(\mathbf{A}^{-1} - \mathbf{B}^{-1} = \mathbf{A}^{-1}(\mathbf{B} - \mathbf{A})\mathbf{B}^{-1}$ for invertible $\mathbf{A}, \mathbf{B})$ that

$$\mathbb{E}[\mathbf{Q}] - \tilde{\mathbf{Q}} = \mathbb{E}\left[ \mathbf{Q} \left( \frac{N}{n} \frac{\mathbf{K}_{\cos}}{1 + \alpha_{\cos}} + \frac{N}{n} \frac{\mathbf{K}_{\sin}}{1 + \alpha_{\sin}} - \frac{1}{n} \mathbf{\Sigma_X^\mathsf{T} \Sigma_X} \right) \right] \tilde{\mathbf{Q}}$$

$$= \mathbb{E}[\mathbf{Q}] \frac{N}{n} \left( \frac{\mathbf{K}_{\cos}}{1 + \alpha_{\cos}} + \frac{\mathbf{K}_{\sin}}{1 + \alpha_{\sin}} \right) \tilde{\mathbf{Q}} - \frac{N}{n} \frac{1}{N} \sum_{i=1}^{N} \mathbb{E}[\mathbf{Q} \mathbf{U}_i \mathbf{U}_i^\mathsf{T}] \tilde{\mathbf{Q}}$$

$$= \mathbb{E}[\mathbf{Q}] \frac{N}{n} \left( \frac{\mathbf{K}_{\cos}}{1 + \alpha_{\cos}} + \frac{\mathbf{K}_{\sin}}{1 + \alpha_{\sin}} \right) \tilde{\mathbf{Q}} - \frac{N}{n} \frac{1}{N} \sum_{i=1}^{N} \mathbb{E}[\mathbf{Q}_{-i} \mathbf{U}_i (\mathbf{I}_2 + \frac{1}{n} \mathbf{U}_i^\mathsf{T} \mathbf{Q}_{-i} \mathbf{U}_i)^{-1} \mathbf{U}_i^\mathsf{T}] \tilde{\mathbf{Q}},$$

for $\mathbf{Q}_{-i} \equiv \left( \frac{1}{n} \mathbf{\Sigma_X^\mathsf{T} \Sigma_X} - \frac{1}{n} \mathbf{U}_i \mathbf{U}_i + \lambda \mathbf{I}_n \right)^{-1}$ that is **independent** of $\mathbf{U}_i$ (and thus $\mathbf{w}_i$), where we applied the following Woodbury identity.

**Lemma 2** (Woodbury). *For* $\mathbf{A}, \mathbf{A} + \mathbf{UU}^\mathsf{T} \in \mathbb{R}^{p \times p}$ *both invertible and* $\mathbf{U} \in \mathbb{R}^{p \times n}$, *we have*

$$(\mathbf{A} + \mathbf{UU}^\mathsf{T})^{-1} = \mathbf{A}^{-1} - \mathbf{A}^{-1} \mathbf{U} (\mathbf{I}_n + \mathbf{U}^\mathsf{T} \mathbf{A}^{-1} \mathbf{U})^{-1} \mathbf{U}^\mathsf{T} \mathbf{A}^{-1}$$

*so that in particular* $(\mathbf{A} + \mathbf{UU}^\mathsf{T})^{-1} \mathbf{U} = \mathbf{A}^{-1} \mathbf{U} (\mathbf{I}_n + \mathbf{U}^\mathsf{T} \mathbf{A}^{-1} \mathbf{U})^{-1}$.

Consider now the two-by-two matrix

$$\mathbf{I}_2 + \frac{1}{n} \mathbf{U}_i^\mathsf{T} \mathbf{Q}_{-i} \mathbf{U}_i = \begin{bmatrix} 1 + \frac{1}{n} \cos(\mathbf{w}_i^\mathsf{T} \mathbf{X}) \mathbf{Q}_{-i} \cos(\mathbf{X^\mathsf{T} w}_i) & \frac{1}{n} \cos(\mathbf{w}_i^\mathsf{T} \mathbf{X}) \mathbf{Q}_{-i} \sin(\mathbf{X^\mathsf{T} w}_i) \\ \frac{1}{n} \sin(\mathbf{w}_i^\mathsf{T} \mathbf{X}) \mathbf{Q}_{-i} \cos(\mathbf{X^\mathsf{T} w}_i) & 1 + \frac{1}{n} \sin(\mathbf{w}_i^\mathsf{T} \mathbf{X}) \mathbf{Q}_{-i} \sin(\mathbf{X^\mathsf{T} w}_i) \end{bmatrix}$$

which, according to the following lemma, is expected to be close to $\begin{bmatrix} 1 + \alpha_{\cos} & 0 \\ 0 & 1 + \alpha_{\sin} \end{bmatrix}$ as defined in (17).

**Lemma 3** (Concentration of quadratic forms). *Under Assumption 1, for* $\sigma_1(\cdot), \sigma_2(\cdot)$ *two real* 1-*Lipschitz functions,* $\mathbf{w} \sim \mathcal{N}(\mathbf{0}, \mathbf{I}_p)$ *and* $\mathbf{A} \in \mathbb{R}^{n \times n}$ *independent of* $\mathbf{w}$ *with* $\|\mathbf{A}\| \leq 1$, *then*

$$\mathbb{P}\left( \left| \frac{1}{n} \sigma_a(\mathbf{w}^\mathsf{T} \mathbf{X}) \mathbf{A} \sigma_b(\mathbf{X^\mathsf{T} w}) - \frac{1}{n} \operatorname{tr}(\mathbf{A} \mathbb{E}_\mathbf{w}[\sigma_b(\mathbf{X^\mathsf{T} w}) \sigma_a(\mathbf{w}^\mathsf{T} \mathbf{X})]) \right| > t \right) \leq C e^{-cn \min(t, t^2)}$$

*for* $a, b \in \{1, 2\}$ *and some universal constants* $C, c > 0$.

*Proof of Lemma 3.* Lemma 3 can be easily extended from [30, Lemma 1], where one observes the proof actually holds when different types of nonlinear Lipschitz functions $\sigma_1(\cdot), \sigma_2(\cdot)$ (and in particular $\cos$ and $\sin$) are considered. □

For $\mathbf{W}_{-i} \in \mathbb{R}^{(N-1) \times p}$ the random matrix $\mathbf{W} \in \mathbb{R}^{N \times p}$ with its $i$-th row $\mathbf{w}_i$ removed, Lemma 3, together with the Lipschitz nature of the map $\mathbf{W}_{-i} \mapsto \frac{1}{n} \sigma_a(\mathbf{w}_i^\mathsf{T} \mathbf{X}) \mathbf{Q}_{-i} \sigma_b(\mathbf{X^\mathsf{T} w}_i)$ for $\mathbf{Q}_{-i} = (\frac{1}{n} \cos(\mathbf{W}_{-i} \mathbf{X})^\mathsf{T} \cos(\mathbf{W}_{-i} \mathbf{X}) + \frac{1}{n} \sin(\mathbf{W}_{-i} \mathbf{X})^\mathsf{T} \sin(\mathbf{W}_{-i} \mathbf{X}) + \lambda \mathbf{I}_n)^{-1}$, leads to the following concentration result

$$\mathbb{P}\left( \left| \frac{1}{n} \sigma_a(\mathbf{w}_i^\mathsf{T} \mathbf{X}) \mathbf{Q}_{-i} \sigma_b(\mathbf{X^\mathsf{T} w}_i) - \frac{1}{n} \operatorname{tr} \left( \mathbb{E}[\mathbf{Q}_{-i}] \mathbb{E}[\sigma_b(\mathbf{X^\mathsf{T} w}_i) \sigma_a(\mathbf{w}_i^\mathsf{T} \mathbf{X})] \right) \right| > t \right) \leq C' e^{-c'n \max(t^2, t)} \tag{18}$$

the proof of which follows the same line of argument of [30, Lemma 4] and is omitted here.

As a consequence, we continue to write, with again the resolvent identity, that

$$(\mathbf{I}_2 + \frac{1}{n}\mathbf{U}_i^\mathsf{T}\mathbf{Q}_{-i}\mathbf{U}_i)^{-1} - \begin{bmatrix} 1+\alpha_{\cos} & 0 \\ 0 & 1+\alpha_{\sin} \end{bmatrix}^{-1}$$

$$= \begin{bmatrix} 1+\frac{1}{n}\cos(\mathbf{w}_i^\mathsf{T}\mathbf{X})\mathbf{Q}_{-i}\cos(\mathbf{X}^\mathsf{T}\mathbf{w}_i) & \frac{1}{n}\cos(\mathbf{w}_i^\mathsf{T}\mathbf{X})\mathbf{Q}_{-i}\sin(\mathbf{X}^\mathsf{T}\mathbf{w}_i) \\ \frac{1}{n}\sin(\mathbf{w}_i^\mathsf{T}\mathbf{X})\mathbf{Q}_{-i}\cos(\mathbf{X}^\mathsf{T}\mathbf{w}_i) & 1+\frac{1}{n}\sin(\mathbf{w}_i^\mathsf{T}\mathbf{X})\mathbf{Q}_{-i}\sin(\mathbf{X}^\mathsf{T}\mathbf{w}_i) \end{bmatrix}^{-1} - \begin{bmatrix} 1+\alpha_{\cos} & 0 \\ 0 & 1+\alpha_{\sin} \end{bmatrix}^{-1}$$

$$= (\mathbf{I}_2 + \frac{1}{n}\mathbf{U}_i^\mathsf{T}\mathbf{Q}_{-i}\mathbf{U}_i)^{-1} \begin{bmatrix} \alpha_{\cos} - \frac{1}{n}\cos(\mathbf{w}_i^\mathsf{T}\mathbf{X})\mathbf{Q}_{-i}\cos(\mathbf{X}^\mathsf{T}\mathbf{w}_i) & -\frac{1}{n}\cos(\mathbf{w}_i^\mathsf{T}\mathbf{X})\mathbf{Q}_{-i}\sin(\mathbf{X}^\mathsf{T}\mathbf{w}_i) \\ -\frac{1}{n}\sin(\mathbf{w}_i^\mathsf{T}\mathbf{X})\mathbf{Q}_{-i}\cos(\mathbf{X}^\mathsf{T}\mathbf{w}_i) & \alpha_{\sin} - \frac{1}{n}\sin(\mathbf{w}_i^\mathsf{T}\mathbf{X})\mathbf{Q}_{-i}\sin(\mathbf{X}^\mathsf{T}\mathbf{w}_i) \end{bmatrix}$$

$$\times \begin{bmatrix} \frac{1}{1+\alpha_{\cos}} & 0 \\ 0 & \frac{1}{1+\alpha_{\sin}} \end{bmatrix} \equiv (\mathbf{I}_2 + \frac{1}{n}\mathbf{U}_i^\mathsf{T}\mathbf{Q}_{-i}\mathbf{U}_i)^{-1}\mathbf{D}_i \begin{bmatrix} \frac{1}{1+\alpha_{\cos}} & 0 \\ 0 & \frac{1}{1+\alpha_{\sin}} \end{bmatrix},$$

where we note from (18) (and $\|\mathbf{Q}_{-i}\| \leq \lambda^{-1}$) that the matrix $\mathbb{E}[\mathbf{D}_i] = o_{\|\cdot\|}(1)$ (in fact of spectral norm of order $O(n^{-\frac{1}{2}})$). So that

$$\mathbb{E}[\mathbf{Q}] - \tilde{\mathbf{Q}} = \mathbb{E}[\mathbf{Q}]\frac{N}{n}\left(\frac{\mathbf{K}_{\cos}}{1+\alpha_{\cos}} + \frac{\mathbf{K}_{\sin}}{1+\alpha_{\sin}}\right)\tilde{\mathbf{Q}} - \frac{N}{n}\frac{1}{N}\sum_{i=1}^{N}\mathbb{E}[\mathbf{Q}_{-i}\mathbf{U}_i(\mathbf{I}_2 + \frac{1}{n}\mathbf{U}_i^\mathsf{T}\mathbf{Q}_{-i}\mathbf{U}_i)^{-1}\mathbf{U}_i^\mathsf{T}]\tilde{\mathbf{Q}}$$

$$= \mathbb{E}[\mathbf{Q}]\frac{N}{n}\left(\frac{\mathbf{K}_{\cos}}{1+\alpha_{\cos}} + \frac{\mathbf{K}_{\sin}}{1+\alpha_{\sin}}\right)\tilde{\mathbf{Q}} - \frac{N}{n}\frac{1}{N}\sum_{i=1}^{N}\mathbb{E}[\mathbf{Q}_{-i}\mathbf{U}_i \begin{bmatrix} \frac{1}{1+\alpha_{\cos}} & 0 \\ 0 & \frac{1}{1+\alpha_{\sin}} \end{bmatrix}\mathbf{U}_i^\mathsf{T}]\tilde{\mathbf{Q}}$$

$$- \frac{N}{n}\frac{1}{N}\sum_{i=1}^{N}\mathbb{E}[\mathbf{Q}_{-i}\mathbf{U}_i(\mathbf{I}_2 + \frac{1}{n}\mathbf{U}_i^\mathsf{T}\mathbf{Q}_{-i}\mathbf{U}_i)^{-1}\mathbf{D}_i \begin{bmatrix} \frac{1}{1+\alpha_{\cos}} & 0 \\ 0 & \frac{1}{1+\alpha_{\sin}} \end{bmatrix}\mathbf{U}_i^\mathsf{T}]\tilde{\mathbf{Q}}$$

$$= (\mathbb{E}[\mathbf{Q}] - \frac{1}{N}\sum_{i=1}^{N}\mathbb{E}[\mathbf{Q}_{-i}])\frac{N}{n}\left(\frac{\mathbf{K}_{\cos}}{1+\alpha_{\cos}} + \frac{\mathbf{K}_{\sin}}{1+\alpha_{\sin}}\right)\tilde{\mathbf{Q}} - \frac{N}{n}\frac{1}{N}\sum_{i=1}^{N}\mathbb{E}[\mathbf{Q}\mathbf{U}_i\mathbf{D}_i \begin{bmatrix} \frac{1}{1+\alpha_{\cos}} & 0 \\ 0 & \frac{1}{1+\alpha_{\sin}} \end{bmatrix}\mathbf{U}_i^\mathsf{T}]\tilde{\mathbf{Q}},$$

where we used $\mathbb{E}_{\mathbf{w}_i}[\mathbf{U}_i\mathbf{U}_i^\mathsf{T}] = \mathbf{K}_{\cos} + \mathbf{K}_{\sin}$ by Lemma 1 and then Lemma 2 in reverse for the last equality. Moreover, since

$$\mathbb{E}[\mathbf{Q}] - \frac{1}{N}\sum_{i=1}^{N}\mathbb{E}[\mathbf{Q}_{-i}] = \frac{1}{N}\sum_{i=1}^{N}\mathbb{E}[\mathbf{Q} - \mathbf{Q}_{-i}] = -\frac{1}{n}\frac{1}{N}\sum_{i=1}^{N}\mathbb{E}[\mathbf{Q}\mathbf{U}_i(\mathbf{I}_2 + \frac{1}{n}\mathbf{U}_i^\mathsf{T}\mathbf{Q}_{-i}\mathbf{U}_i)^{-1}\mathbf{U}_i^\mathsf{T}\mathbf{Q}]$$

so that with the fact $\frac{1}{\sqrt{n}}\|\mathbf{Q}\boldsymbol{\Sigma}_\mathbf{X}^\mathsf{T}\| \leq \|\sqrt{\mathbf{Q}\frac{1}{n}\boldsymbol{\Sigma}_\mathbf{X}^\mathsf{T}\boldsymbol{\Sigma}_\mathbf{X}\mathbf{Q}}\| \leq \lambda^{-\frac{1}{2}}$ we have for the first term

$$\|\mathbb{E}[\mathbf{Q}] - \frac{1}{N}\sum_{i=1}^{N}\mathbb{E}[\mathbf{Q}_{-i}]\| = O(n^{-1}).$$

It thus remains to treat the second term, which, with the relation $\mathbf{A}\mathbf{B}^\mathsf{T} + \mathbf{B}\mathbf{A}^\mathsf{T} \preceq \mathbf{A}\mathbf{A}^\mathsf{T} + \mathbf{B}\mathbf{B}^\mathsf{T}$ (in the sense of symmetric matrices), and the same line of arguments as above, can be shown to have vanishing spectral norm (of order $O(n^{-\frac{1}{2}})$) as $n, p, N \to \infty$.

We thus have $\|\mathbb{E}[\mathbf{Q}] - \tilde{\mathbf{Q}}\| = O(n^{-\frac{1}{2}})$, which concludes the first part of the proof of Theorem 1.

We shall show next that $\|\tilde{\mathbf{Q}} - \bar{\mathbf{Q}}\| \to 0$ as $n, p, N \to \infty$. First note from previous derivation that $\alpha_\sigma - \frac{1}{n}\operatorname{tr}\mathbf{K}_\sigma\tilde{\mathbf{Q}} = O(n^{-\frac{1}{2}})$ for $\sigma = \cos, \sin$. To compare $\tilde{\mathbf{Q}}$ and $\bar{\mathbf{Q}}$, it follows again from the resolvent identity that

$$\tilde{\mathbf{Q}} - \bar{\mathbf{Q}} = \tilde{\mathbf{Q}}\left(\frac{N}{n}\frac{\mathbf{K}_{\cos}(\alpha_{\cos} - \delta_{\cos})}{(1+\delta_{\cos})(1+\alpha_{\cos})} + \frac{N}{n}\frac{\mathbf{K}_{\sin}(\alpha_{\sin} - \delta_{\sin})}{(1+\delta_{\sin})(1+\alpha_{\sin})}\right)\bar{\mathbf{Q}}$$

so that the control of $\|\tilde{\mathbf{Q}} - \bar{\mathbf{Q}}\|$ boils down to the control of $\max\{|\alpha_{\cos} - \delta_{\cos}|, |\alpha_{\sin} - \delta_{\sin}|\}$. To this end, it suffices to write

$$\alpha_{\cos} - \delta_{\cos} = \frac{1}{n}\operatorname{tr}\mathbf{K}_{\cos}(\mathbb{E}[\mathbf{Q}] - \bar{\mathbf{Q}}) = \frac{1}{n}\operatorname{tr}\mathbf{K}_{\cos}(\tilde{\mathbf{Q}} - \bar{\mathbf{Q}}) + O(n^{-\frac{1}{2}})$$

where we used $|\operatorname{tr}(\mathbf{AB})| \le \|\mathbf{A}\| \operatorname{tr}(\mathbf{B})$ for nonnegative definite $\mathbf{B}$, together with the fact that $\frac{1}{n}\operatorname{tr}\mathbf{K}_\sigma$ is (uniformly) bounded under Assumption 1, for $\sigma = \cos, \sin$.

As a consequence, we have

$$|\alpha_{\cos} - \delta_{\cos}| \le |\alpha_{\cos} - \delta_{\cos}| \frac{N}{n} \frac{\frac{1}{n}\operatorname{tr}(\mathbf{K}_{\cos}\tilde{\mathbf{Q}}\mathbf{K}_{\cos}\bar{\mathbf{Q}})}{(1+\delta_{\cos})(1+\alpha_{\cos})} + o(1).$$

It thus remains to show

$$\frac{N}{n}\frac{\frac{1}{n}\operatorname{tr}(\mathbf{K}_{\cos}\tilde{\mathbf{Q}}\mathbf{K}_{\cos}\bar{\mathbf{Q}})}{(1+\delta_{\cos})(1+\alpha_{\cos})} < 1$$

or alternatively, by the Cauchy–Schwarz inequality, to show

$$\frac{N}{n}\frac{\frac{1}{n}\operatorname{tr}(\mathbf{K}_{\cos}\tilde{\mathbf{Q}}\mathbf{K}_{\cos}\bar{\mathbf{Q}})}{(1+\delta_{\cos})(1+\alpha_{\cos})} \le \sqrt{\frac{N}{n}\frac{\frac{1}{n}\operatorname{tr}(\mathbf{K}_{\cos}\bar{\mathbf{Q}}\mathbf{K}_{\cos}\bar{\mathbf{Q}})}{(1+\delta_{\cos})^2} \cdot \frac{N}{n}\frac{\frac{1}{n}\operatorname{tr}(\mathbf{K}_{\cos}\tilde{\mathbf{Q}}\mathbf{K}_{\cos}\tilde{\mathbf{Q}})}{(1+\alpha_{\cos})^2}} < 1.$$

To treat the first right-hand side term (the second can be done similarly), it unfolds from $|\operatorname{tr}(\mathbf{AB})| \le \|\mathbf{A}\| \cdot \operatorname{tr}(\mathbf{B})$ for nonnegative definite $\mathbf{B}$ that

$$\frac{N}{n}\frac{\frac{1}{n}\operatorname{tr}(\mathbf{K}_{\cos}\bar{\mathbf{Q}}\mathbf{K}_{\cos}\bar{\mathbf{Q}})}{(1+\delta_{\cos})^2} \le \left\|\frac{N}{n}\frac{\mathbf{K}_{\cos}\bar{\mathbf{Q}}}{1+\delta_{\cos}}\right\| \frac{\frac{1}{n}\operatorname{tr}(\mathbf{K}_{\cos}\bar{\mathbf{Q}})}{1+\delta_{\cos}} = \left\|\frac{N}{n}\frac{\mathbf{K}_{\cos}\bar{\mathbf{Q}}}{1+\delta_{\cos}}\right\| \frac{\gamma_{\cos}}{1+\delta_{\cos}} \le \frac{\gamma_{\cos}}{1+\delta_{\cos}} < 1$$

where we used the fact that $\frac{N}{n}\frac{\mathbf{K}_{\cos}\bar{\mathbf{Q}}}{1+\delta_{\cos}} = \mathbf{I}_n - \frac{N}{n}\frac{\mathbf{K}_{\sin}\bar{\mathbf{Q}}}{1+\delta_{\sin}} - \lambda\bar{\mathbf{Q}}$. This concludes the proof of Theorem 1.
∎

## C   Proof of Theorem 2

To prove Theorem 2, it indeed suffices to prove the following lemma.

**Lemma 4** (Asymptotic behavior of $\mathbb{E}[\mathbf{QAQ}]$). *Under Assumption 1, for $\mathbf{Q}$ defined in (5) and symmetric nonnegative definite $\mathbf{A} \in \mathbb{R}^{n\times n}$ of bounded spectral norm, we have*

$$\left\|\mathbb{E}[\mathbf{QAQ}] - \left(\bar{\mathbf{Q}}\mathbf{A}\bar{\mathbf{Q}} + \frac{N}{n}\left[\frac{\frac{1}{n}\operatorname{tr}(\bar{\mathbf{Q}}\mathbf{A}\bar{\mathbf{Q}}\mathbf{K}_{\cos})}{(1+\delta_{\cos})^2} \quad \frac{\frac{1}{n}\operatorname{tr}(\bar{\mathbf{Q}}\mathbf{A}\bar{\mathbf{Q}}\mathbf{K}_{\sin})}{(1+\delta_{\sin})^2}\right]\mathbf{\Omega}\begin{bmatrix}\bar{\mathbf{Q}}\mathbf{K}_{\cos}\bar{\mathbf{Q}}\\\bar{\mathbf{Q}}\mathbf{K}_{\sin}\bar{\mathbf{Q}}\end{bmatrix}\right)\right\| \to 0$$

*almost surely as $n \to \infty$, with $\mathbf{\Omega}^{-1} \equiv \mathbf{I}_2 - \frac{N}{n}\begin{bmatrix}\frac{\frac{1}{n}\operatorname{tr}(\bar{\mathbf{Q}}\mathbf{K}_{\cos}\bar{\mathbf{Q}}\mathbf{K}_{\cos})}{(1+\delta_{\cos})^2} & \frac{\frac{1}{n}\operatorname{tr}(\bar{\mathbf{Q}}\mathbf{K}_{\cos}\bar{\mathbf{Q}}\mathbf{K}_{\sin})}{(1+\delta_{\sin})^2}\\\frac{\frac{1}{n}\operatorname{tr}(\bar{\mathbf{Q}}\mathbf{K}_{\cos}\bar{\mathbf{Q}}\mathbf{K}_{\sin})}{(1+\delta_{\cos})^2} & \frac{\frac{1}{n}\operatorname{tr}(\bar{\mathbf{Q}}\mathbf{K}_{\sin}\bar{\mathbf{Q}}\mathbf{K}_{\sin})}{(1+\delta_{\sin})^2}\end{bmatrix}$. In particular, we have*

$$\left\|\mathbb{E}\begin{bmatrix}\mathbf{Q}\mathbf{K}_{\cos}\mathbf{Q}\\\mathbf{Q}\mathbf{K}_{\sin}\mathbf{Q}\end{bmatrix} - \mathbf{\Omega}\begin{bmatrix}\bar{\mathbf{Q}}\mathbf{K}_{\cos}\bar{\mathbf{Q}}\\\bar{\mathbf{Q}}\mathbf{K}_{\sin}\bar{\mathbf{Q}}\end{bmatrix}\right\| \to 0.$$

*Proof of Lemma 4.* The proof of Lemma 4 essentially follows the same line of arguments as that of Theorem 1. Writing

$$\begin{aligned}
\mathbb{E}[\mathbf{QAQ}] &= \mathbb{E}[\bar{\mathbf{Q}}\mathbf{AQ}] + \mathbb{E}[(\mathbf{Q}-\bar{\mathbf{Q}})\mathbf{AQ}]\\
&\simeq \bar{\mathbf{Q}}\mathbf{A}\bar{\mathbf{Q}} + \mathbb{E}\left[\mathbf{Q}\left(\frac{N}{n}\frac{\mathbf{K}_{\cos}}{1+\delta_{\cos}} + \frac{N}{n}\frac{\mathbf{K}_{\sin}}{1+\delta_{\sin}} - \frac{1}{n}\mathbf{\Sigma}_{\mathbf{X}}^\mathsf{T}\mathbf{\Sigma}_{\mathbf{X}}\right)\bar{\mathbf{Q}}\mathbf{A}\mathbf{Q}\right]\\
&= \bar{\mathbf{Q}}\mathbf{A}\bar{\mathbf{Q}} + \frac{N}{n}\mathbb{E}[\mathbf{Q}\mathbf{\Phi}\bar{\mathbf{Q}}\mathbf{A}\mathbf{Q}] - \frac{1}{n}\sum_{i=1}^N \mathbb{E}[\mathbf{Q}\mathbf{U}_i\mathbf{U}_i^\mathsf{T}\bar{\mathbf{Q}}\mathbf{A}\mathbf{Q}]
\end{aligned}$$

where we note $\simeq$ by ignoring matrices with vanishing spectral norm (i.e., $o_{\|\cdot\|}(1)$) in the $n, , p, N \to \infty$ limit and recall the shortcut $\boldsymbol{\Phi} \equiv \frac{\mathbf{K}_{\cos}}{1+\delta_{\cos}} + \frac{\mathbf{K}_{\sin}}{1+\delta_{\sin}}$. Developing rightmost term with Lemma 2 as

$$\mathbb{E}[\mathbf{Q}\mathbf{U}_i\mathbf{U}_i^\mathsf{T}\bar{\mathbf{Q}}\mathbf{A}\mathbf{Q}] = \mathbb{E}\left[\mathbf{Q}_{-i}\mathbf{U}_i(\mathbf{I}_2 + \frac{1}{n}\mathbf{U}_i^\mathsf{T}\mathbf{Q}_{-i}\mathbf{U}_i)^{-1}\mathbf{U}_i^\mathsf{T}\bar{\mathbf{Q}}\mathbf{A}\mathbf{Q}\right]$$

$$= \mathbb{E}\left[\mathbf{Q}_{-i}\mathbf{U}_i(\mathbf{I}_2 + \frac{1}{n}\mathbf{U}_i^\mathsf{T}\mathbf{Q}_{-i}\mathbf{U}_i)^{-1}\mathbf{U}_i^\mathsf{T}\bar{\mathbf{Q}}\mathbf{A}\mathbf{Q}_{-i}\right]$$

$$- \frac{1}{n}\mathbb{E}\left[\mathbf{Q}_{-i}\mathbf{U}_i(\mathbf{I}_2 + \frac{1}{n}\mathbf{U}_i^\mathsf{T}\mathbf{Q}_{-i}\mathbf{U}_i)^{-1}\mathbf{U}_i^\mathsf{T}\bar{\mathbf{Q}}\mathbf{A}\mathbf{Q}_{-i}\mathbf{U}_i(\mathbf{I}_2 + \frac{1}{n}\mathbf{U}_i^\mathsf{T}\mathbf{Q}_{-i}\mathbf{U}_i)^{-1}\mathbf{U}_i^\mathsf{T}\mathbf{Q}_{-i}\right]$$

$$\simeq \mathbb{E}[\mathbf{Q}_{-i}\boldsymbol{\Phi}\bar{\mathbf{Q}}\mathbf{A}\mathbf{Q}_{-i}]$$

$$- \mathbb{E}\left[\mathbf{Q}_{-i}\mathbf{U}_i\begin{bmatrix}\frac{1}{1+\delta_{\cos}} & 0 \\ 0 & \frac{1}{1+\delta_{\sin}}\end{bmatrix}\begin{bmatrix}\frac{1}{n}\operatorname{tr}(\bar{\mathbf{Q}}\mathbf{A}\bar{\mathbf{Q}}\mathbf{K}_{\cos}) & 0 \\ 0 & \frac{1}{n}\operatorname{tr}(\bar{\mathbf{Q}}\mathbf{A}\bar{\mathbf{Q}}\mathbf{K}_{\sin})\end{bmatrix}\begin{bmatrix}\frac{1}{1+\delta_{\cos}} & 0 \\ 0 & \frac{1}{1+\delta_{\sin}}\end{bmatrix}\mathbf{U}_i^\mathsf{T}\mathbf{Q}_{-i}\right]$$

so that

$$\mathbb{E}[\mathbf{Q}\mathbf{A}\mathbf{Q}] \simeq \bar{\mathbf{Q}}\mathbf{A}\bar{\mathbf{Q}} + \frac{N}{n}\mathbb{E}\left[\mathbf{Q}\left(\frac{\frac{1}{n}\operatorname{tr}(\bar{\mathbf{Q}}\mathbf{A}\bar{\mathbf{Q}}\mathbf{K}_{\cos})}{(1+\delta_{\cos})^2}\mathbf{K}_{\cos} + \frac{\frac{1}{n}\operatorname{tr}(\bar{\mathbf{Q}}\mathbf{A}\bar{\mathbf{Q}}\mathbf{K}_{\sin})}{(1+\delta_{\sin})^2}\mathbf{K}_{\sin}\right)\mathbf{Q}\right]$$

$$= \bar{\mathbf{Q}}\mathbf{A}\bar{\mathbf{Q}} + \frac{N}{n}\begin{bmatrix}\frac{\frac{1}{n}\operatorname{tr}(\bar{\mathbf{Q}}\mathbf{A}\bar{\mathbf{Q}}\mathbf{K}_{\cos})}{(1+\delta_{\cos})^2} & \frac{\frac{1}{n}\operatorname{tr}(\bar{\mathbf{Q}}\mathbf{A}\bar{\mathbf{Q}}\mathbf{K}_{\sin})}{(1+\delta_{\sin})^2}\end{bmatrix}\mathbb{E}\begin{bmatrix}\mathbf{Q}\mathbf{K}_{\cos}\mathbf{Q} \\ \mathbf{Q}\mathbf{K}_{\sin}\mathbf{Q}\end{bmatrix} \qquad (19)$$

by taking $\mathbf{A} = \mathbf{K}_{\cos}$ or $\mathbf{K}_{\sin}$, we result in

$$\mathbb{E}[\mathbf{Q}\mathbf{K}_{\cos}\mathbf{Q}] \simeq \frac{c}{ac-bd}\bar{\mathbf{Q}}\mathbf{K}_{\cos}\bar{\mathbf{Q}} + \frac{b}{ac-bd}\bar{\mathbf{Q}}\mathbf{K}_{\sin}\bar{\mathbf{Q}}$$

$$\mathbb{E}[\mathbf{Q}\mathbf{K}_{\sin}\mathbf{Q}] \simeq \frac{a}{ac-bd}\bar{\mathbf{Q}}\mathbf{K}_{\sin}\bar{\mathbf{Q}} + \frac{d}{ac-bd}\bar{\mathbf{Q}}\mathbf{K}_{\cos}\bar{\mathbf{Q}}$$

with $a = 1 - \frac{N}{n}\frac{\frac{1}{n}\operatorname{tr}(\bar{\mathbf{Q}}\mathbf{K}_{\cos}\bar{\mathbf{Q}}\mathbf{K}_{\cos})}{(1+\delta_{\cos})^2}$, $b = \frac{N}{n}\frac{\frac{1}{n}\operatorname{tr}(\bar{\mathbf{Q}}\mathbf{K}_{\cos}\bar{\mathbf{Q}}\mathbf{K}_{\sin})}{(1+\delta_{\sin})^2}$, $c = 1 - \frac{N}{n}\frac{\frac{1}{n}\operatorname{tr}(\bar{\mathbf{Q}}\mathbf{K}_{\sin}\bar{\mathbf{Q}}\mathbf{K}_{\sin})}{(1+\delta_{\sin})^2}$ and $d = \frac{N}{n}\frac{\frac{1}{n}\operatorname{tr}(\bar{\mathbf{Q}}\mathbf{K}_{\sin}\bar{\mathbf{Q}}\mathbf{K}_{\cos})}{(1+\delta_{\cos})^2}$ such that $(1+\delta_{\sin})^2 b = (1+\delta_{\cos})^2 d$.

$$\mathbb{E}\begin{bmatrix}\mathbf{Q}\mathbf{K}_{\cos}\mathbf{Q} \\ \mathbf{Q}\mathbf{K}_{\sin}\mathbf{Q}\end{bmatrix} \simeq \begin{bmatrix}a & -b \\ -d & c\end{bmatrix}^{-1}\begin{bmatrix}\bar{\mathbf{Q}}\mathbf{K}_{\cos}\bar{\mathbf{Q}} \\ \bar{\mathbf{Q}}\mathbf{K}_{\sin}\bar{\mathbf{Q}}\end{bmatrix} \equiv \boldsymbol{\Omega}\begin{bmatrix}\bar{\mathbf{Q}}\mathbf{K}_{\cos}\bar{\mathbf{Q}} \\ \bar{\mathbf{Q}}\mathbf{K}_{\sin}\bar{\mathbf{Q}}\end{bmatrix}$$

for $\boldsymbol{\Omega} \equiv \begin{bmatrix}a & -b \\ -d & c\end{bmatrix}^{-1}$. Plugging back into (19) we conclude the proof of Lemma 4. $\square$

Theorem 2 can be achieved by considering the concentration of (the bilinear form) $\frac{1}{n}\mathbf{y}^\mathsf{T}\mathbf{Q}^2\mathbf{y}$ around its expectation $\frac{1}{n}\mathbf{y}^\mathsf{T}\mathbb{E}[\mathbf{Q}^2]\mathbf{y}$ (with for instance Lemma 3 in [30]), together with Lemma 4. This concludes the proof of Theorem 2. $\blacksquare$

# D  Proof of Theorem 3

Recall the definition of $E_{\text{test}} = \frac{1}{\hat{n}}\|\hat{\mathbf{y}} - \boldsymbol{\Sigma}_{\hat{\mathbf{X}}}^\mathsf{T}\boldsymbol{\beta}\|^2$ from (4) with $\boldsymbol{\Sigma}_{\hat{\mathbf{X}}} = \begin{bmatrix}\cos(\mathbf{W}\hat{\mathbf{X}}) \\ \sin(\mathbf{W}\hat{\mathbf{X}})\end{bmatrix} \in \mathbb{R}^{2N\times\hat{n}}$ on a test set $(\hat{\mathbf{X}}, \hat{\mathbf{y}})$ of size $\hat{n}$, and first focus on the case $2N > n$ where $\boldsymbol{\beta} = \frac{1}{n}\boldsymbol{\Sigma}_{\mathbf{X}}\mathbf{Q}\mathbf{y}$ as per (3). By (15), we have

$$E_{\text{test}} = \frac{1}{\hat{n}}\left\|\hat{\mathbf{y}} - \frac{1}{n}\boldsymbol{\Sigma}_{\hat{\mathbf{X}}}^\mathsf{T}\boldsymbol{\Sigma}_{\mathbf{X}}\mathbf{Q}\mathbf{y}\right\|^2 = \frac{1}{\hat{n}}\left\|\hat{\mathbf{y}} - \frac{1}{n}\sum_{i=1}^{N}\hat{\mathbf{U}}_i\mathbf{U}_i^\mathsf{T}\mathbf{Q}\mathbf{y}\right\|^2$$

where, similar to the notation $\mathbf{U}_i = \begin{bmatrix}\cos(\mathbf{X}^\mathsf{T}\mathbf{w}_i) & \sin(\mathbf{X}^\mathsf{T}\mathbf{w}_i)\end{bmatrix} \in \mathbb{R}^{n\times2}$ as in the proof of Theorem 1, we denote

$$\hat{\mathbf{U}}_i \equiv \begin{bmatrix}\cos(\hat{\mathbf{X}}^\mathsf{T}\mathbf{w}_i) & \sin(\hat{\mathbf{X}}^\mathsf{T}\mathbf{w}_i)\end{bmatrix} \in \mathbb{R}^{\hat{n}\times2}.$$

As a consequence, we further get

$$\mathbb{E}[E_{\text{test}}] = \frac{1}{\hat{n}}\|\hat{\mathbf{y}}\|^2 - \frac{2}{n\hat{n}}\sum_{i=1}^{N}\hat{\mathbf{y}}^{\mathsf{T}}\mathbb{E}[\hat{\mathbf{U}}_i\mathbf{U}_i^{\mathsf{T}}\mathbf{Q}]\mathbf{y} + \frac{1}{n^2\hat{n}}\sum_{i,j=1}^{N}\mathbf{y}^{\mathsf{T}}\mathbb{E}[\mathbf{Q}\mathbf{U}_i\hat{\mathbf{U}}_i^{\mathsf{T}}\hat{\mathbf{U}}_j\mathbf{U}_j^{\mathsf{T}}\mathbf{Q}]\mathbf{y}$$

$$= \frac{1}{\hat{n}}\|\hat{\mathbf{y}}\|^2 - \frac{2}{n\hat{n}}\sum_{i=1}^{N}\hat{\mathbf{y}}^{\mathsf{T}}\mathbb{E}\left[\hat{\mathbf{U}}_i(\mathbf{I}_2 + \frac{1}{n}\mathbf{U}_i^{\mathsf{T}}\mathbf{Q}_{-i}\mathbf{U}_i)^{-1}\mathbf{U}_i^{\mathsf{T}}\mathbf{Q}_{-i}\right]\mathbf{y} + \frac{1}{n^2\hat{n}}\sum_{i,j=1}^{N}\mathbf{y}^{\mathsf{T}}\mathbb{E}[\mathbf{Q}\mathbf{U}_i\hat{\mathbf{U}}_i^{\mathsf{T}}\hat{\mathbf{U}}_j\mathbf{U}_j^{\mathsf{T}}\mathbf{Q}]\mathbf{y}$$

$$\simeq \frac{1}{\hat{n}}\|\hat{\mathbf{y}}\|^2 - \frac{2}{n\hat{n}}\sum_{i=1}^{N}\hat{\mathbf{y}}^{\mathsf{T}}\mathbb{E}\left[\hat{\mathbf{U}}_i\begin{bmatrix}\frac{1}{1+\delta_{\cos}} & 0 \\ 0 & \frac{1}{1+\delta_{\sin}}\end{bmatrix}\mathbf{U}_i^{\mathsf{T}}\mathbf{Q}_{-i}\right]\mathbf{y} + \frac{1}{n^2\hat{n}}\sum_{i,j=1}^{N}\mathbf{y}^{\mathsf{T}}\mathbb{E}[\mathbf{Q}\mathbf{U}_i\hat{\mathbf{U}}_i^{\mathsf{T}}\hat{\mathbf{U}}_j\mathbf{U}_j^{\mathsf{T}}\mathbf{Q}]\mathbf{y}$$

$$\simeq \frac{1}{\hat{n}}\|\hat{\mathbf{y}}\|^2 - \frac{2}{\hat{n}}\hat{\mathbf{y}}^{\mathsf{T}}\left(\frac{N}{n}\frac{\mathbf{K}_{\cos}(\hat{\mathbf{X}},\mathbf{X})}{1+\delta_{\cos}} + \frac{N}{n}\frac{\mathbf{K}_{\sin}(\hat{\mathbf{X}},\mathbf{X})}{1+\delta_{\sin}}\right)\bar{\mathbf{Q}}\mathbf{y} + \frac{1}{n^2\hat{n}}\sum_{i,j=1}^{N}\mathbf{y}^{\mathsf{T}}\mathbb{E}[\mathbf{Q}\mathbf{U}_i\hat{\mathbf{U}}_i^{\mathsf{T}}\hat{\mathbf{U}}_j\mathbf{U}_j^{\mathsf{T}}\mathbf{Q}]\mathbf{y}$$

where we similarly denote

$$\mathbf{K}_{\cos}(\hat{\mathbf{X}},\mathbf{X}) \equiv \left\{e^{-\frac{1}{2}(\|\hat{\mathbf{x}}_i\|^2+\|\mathbf{x}_j\|^2)}\cosh(\hat{\mathbf{x}}_i^{\mathsf{T}}\mathbf{x}_j)\right\}_{i,j=1}^{\hat{n},n}$$

$$\mathbf{K}_{\sin}(\hat{\mathbf{X}},\mathbf{X}) \equiv \left\{e^{-\frac{1}{2}(\|\hat{\mathbf{x}}_i\|^2+\|\mathbf{x}_j\|^2)}\sinh(\hat{\mathbf{x}}_i^{\mathsf{T}}\mathbf{x}_j)\right\}_{i,j=1}^{\hat{n},n} \in \mathbb{R}^{\hat{n}\times n}.$$

Note that, different from the proof of Theorem 1 and 2 where we constantly use the fact that $\|\mathbf{Q}\| \leq \lambda^{-1}$ and

$$\frac{1}{n}\boldsymbol{\Sigma}_{\mathbf{X}}^{\mathsf{T}}\boldsymbol{\Sigma}_{\mathbf{X}}\mathbf{Q} = \mathbf{I}_n - \lambda\mathbf{Q}$$

so that $\|\frac{1}{n}\boldsymbol{\Sigma}_{\mathbf{X}}^{\mathsf{T}}\boldsymbol{\Sigma}_{\mathbf{X}}\mathbf{Q}\| \leq 1$, we do not have in general a simple control for $\|\frac{1}{n}\boldsymbol{\Sigma}_{\hat{\mathbf{X}}}^{\mathsf{T}}\boldsymbol{\Sigma}_{\mathbf{X}}\mathbf{Q}\|$, when arbitrary $\hat{\mathbf{X}}$ is considered. Intuitively speaking, this is due to the loss-of-control for $\|\frac{1}{n}(\boldsymbol{\Sigma}_{\hat{\mathbf{X}}} - \boldsymbol{\Sigma}_{\mathbf{X}})^{\mathsf{T}}\boldsymbol{\Sigma}_{\mathbf{X}}\mathbf{Q}\|$ when $\hat{\mathbf{X}}$ can be chosen arbitrarily with respect to $\mathbf{X}$. It was remarked in [30, Remark 1] that in general only a $O(\sqrt{n})$ upper bound can be derived for $\|\frac{1}{\sqrt{n}}\boldsymbol{\Sigma}_{\mathbf{X}}\|$ or $\|\frac{1}{\sqrt{n}}\boldsymbol{\Sigma}_{\hat{\mathbf{X}}}\|$. Nonetheless, this problem can be resolved with the additional Assumption 2.

More precisely, note that

$$\left\|\frac{1}{n}\boldsymbol{\Sigma}_{\hat{\mathbf{X}}}^{\mathsf{T}}\boldsymbol{\Sigma}_{\mathbf{X}}\mathbf{Q}\right\| \leq \frac{1}{n}\|\boldsymbol{\Sigma}_{\mathbf{X}}^{\mathsf{T}}\boldsymbol{\Sigma}_{\mathbf{X}}\mathbf{Q}\| + \frac{1}{n}\|(\boldsymbol{\Sigma}_{\hat{\mathbf{X}}} - \boldsymbol{\Sigma}_{\mathbf{X}})^{\mathsf{T}}\boldsymbol{\Sigma}_{\mathbf{X}}\mathbf{Q}\| \leq 1 + \frac{1}{\sqrt{n}}\|\boldsymbol{\Sigma}_{\hat{\mathbf{X}}} - \boldsymbol{\Sigma}_{\mathbf{X}}\| \cdot \frac{1}{\sqrt{n}}\|\boldsymbol{\Sigma}_{\mathbf{X}}\mathbf{Q}\| \tag{20}$$

it remains to show that $\|\boldsymbol{\Sigma}_{\mathbf{X}} - \boldsymbol{\Sigma}_{\hat{\mathbf{X}}}\| = O(\sqrt{n})$ under Assumption 2 to establish $\|\frac{1}{n}\boldsymbol{\Sigma}_{\hat{\mathbf{X}}}^{\mathsf{T}}\boldsymbol{\Sigma}_{\mathbf{X}}\mathbf{Q}\| = O(1)$, that is, to show that

$$\|\sigma(\mathbf{W}\mathbf{X}) - \sigma(\mathbf{W}\hat{\mathbf{X}})\| = O(\sqrt{n}) \tag{21}$$

for $\sigma \in \{\cos,\sin\}$. Note this cannot be achieved using only the Lipschitz nature of $\sigma(\cdot)$ and the fact that $\|\mathbf{X} - \hat{\mathbf{X}}\| \leq \|\mathbf{X}\| + \|\hat{\mathbf{X}}\| = O(1)$ under Assumption 1 by writing

$$\|\sigma(\mathbf{W}\mathbf{X}) - \sigma(\mathbf{W}\hat{\mathbf{X}})\| \leq \|\sigma(\mathbf{W}\mathbf{X}) - \sigma(\mathbf{W}\hat{\mathbf{X}})\|_F \leq \|\mathbf{W}\|_F \cdot \|\mathbf{X} - \hat{\mathbf{X}}\| = O(n). \tag{22}$$

where we recall that $\|\mathbf{W}\| = O(\sqrt{n})$ and $\|\mathbf{W}\|_F = O(n)$. Nonetheless, from [29, Proposition B.1] we have that the product $\mathbf{W}\mathbf{X}$, and thus $\sigma(\mathbf{W}\mathbf{X})$, strongly concentrates around its expectation in the sense of (9), so that

$$\|\sigma(\mathbf{W}\mathbf{X}) - \sigma(\mathbf{W}\hat{\mathbf{X}})\| \leq \|\sigma(\mathbf{W}\mathbf{X}) - \mathbb{E}[\sigma(\mathbf{W}\mathbf{X})]\| + \|\mathbb{E}[\sigma(\mathbf{W}\mathbf{X}) - \sigma(\mathbf{W}\hat{\mathbf{X}})]\|$$
$$+ \|\sigma(\mathbf{W}\hat{\mathbf{X}}) - \mathbb{E}[\sigma(\mathbf{W}\hat{\mathbf{X}})]\| = O(\sqrt{n})$$

under Assumption 2. As a results, we are allowed to control $\frac{1}{n}\boldsymbol{\Sigma}_{\hat{\mathbf{X}}}^{\mathsf{T}}\boldsymbol{\Sigma}_{\mathbf{X}}\mathbf{Q}$ and similarly $\frac{1}{n}\boldsymbol{\Sigma}_{\hat{\mathbf{X}}}^{\mathsf{T}}\boldsymbol{\Sigma}_{\hat{\mathbf{X}}}\mathbf{Q}$ in the same vein as $\frac{1}{n}\boldsymbol{\Sigma}_{\mathbf{X}}^{\mathsf{T}}\boldsymbol{\Sigma}_{\mathbf{X}}\mathbf{Q}$ in the proof of Theorem 1 and 2 in Appendix B and C, respectively.

It thus remains to handle the last term (noted $\mathbf{Z}$) as follows

$$\mathbf{Z} \equiv \frac{1}{n^2\hat{n}}\sum_{i,j=1}^{N}\mathbf{y}^{\mathsf{T}}\mathbb{E}[\mathbf{Q}\mathbf{U}_i\hat{\mathbf{U}}_i^{\mathsf{T}}\hat{\mathbf{U}}_j\mathbf{U}_j^{\mathsf{T}}\mathbf{Q}]\mathbf{y}$$

$$= \frac{1}{n^2\hat{n}}\sum_{i=1}^{N}\mathbf{y}^{\mathsf{T}}\mathbb{E}[\mathbf{Q}\mathbf{U}_i\hat{\mathbf{U}}_i^{\mathsf{T}}\hat{\mathbf{U}}_i\mathbf{U}_i^{\mathsf{T}}\mathbf{Q}]\mathbf{y} + \frac{1}{n^2\hat{n}}\sum_{i=1}^{N}\sum_{j\neq i}\mathbf{y}^{\mathsf{T}}\mathbb{E}[\mathbf{Q}\mathbf{U}_i\hat{\mathbf{U}}_i^{\mathsf{T}}\hat{\mathbf{U}}_j\mathbf{U}_j^{\mathsf{T}}\mathbf{Q}]\mathbf{y} = \mathbf{Z}_1 + \mathbf{Z}_2$$

where $\mathbf{Z}_1$ term can be treated as

$$\mathbf{Z}_1 \equiv \frac{1}{n^2 \hat{n}} \sum_{i=1}^N \mathbf{y}^\mathsf{T} \mathbb{E}[\mathbf{Q}\mathbf{U}_i \hat{\mathbf{U}}_i^\mathsf{T} \hat{\mathbf{U}}_i \mathbf{U}_i^\mathsf{T} \mathbf{Q}]\mathbf{y}$$

$$= \frac{1}{n\hat{n}} \sum_{i=1}^N \mathbf{y}^\mathsf{T} \mathbb{E}[\mathbf{Q}_{-i}\mathbf{U}_i(\mathbf{I}_2 + \tfrac{1}{n}\mathbf{U}_i^\mathsf{T}\mathbf{Q}_{-i}\mathbf{U}_i)^{-1}\tfrac{1}{n}\hat{\mathbf{U}}_i^\mathsf{T}\hat{\mathbf{U}}_i(\mathbf{I}_2 + \tfrac{1}{n}\mathbf{U}_i^\mathsf{T}\mathbf{Q}_{-i}\mathbf{U}_i)^{-1}\mathbf{U}_i^\mathsf{T}\mathbf{Q}_{-i}]\mathbf{y}$$

$$\simeq \frac{1}{n\hat{n}} \sum_{i=1}^N \mathbf{y}^\mathsf{T} \mathbb{E}\left[\mathbf{Q}_{-i}\mathbf{U}_i \begin{bmatrix} \frac{1}{1+\delta_{\cos}} & 0 \\ 0 & \frac{1}{1+\delta_{\sin}} \end{bmatrix} \begin{bmatrix} \frac{1}{n}\operatorname{tr}\hat{\hat{\mathbf{K}}}_{\cos} & 0 \\ 0 & \frac{1}{n}\operatorname{tr}\hat{\hat{\mathbf{K}}}_{\sin} \end{bmatrix} \begin{bmatrix} \frac{1}{1+\delta_{\cos}} & 0 \\ 0 & \frac{1}{1+\delta_{\sin}} \end{bmatrix} \mathbf{U}_i^\mathsf{T}\mathbf{Q}_{-i}\right]\mathbf{y}$$

$$\simeq \frac{N}{n}\frac{1}{\hat{n}}\mathbf{y}^\mathsf{T}\mathbb{E}\left[\mathbf{Q}\left(\frac{\frac{1}{n}\operatorname{tr}\mathbf{K}_{\cos}(\hat{\mathbf{X}},\hat{\mathbf{X}})}{(1+\delta_{\cos})^2}\mathbf{K}_{\cos} + \frac{\frac{1}{n}\operatorname{tr}\mathbf{K}_{\sin}(\hat{\mathbf{X}},\hat{\mathbf{X}})}{(1+\delta_{\sin})^2}\mathbf{K}_{\sin}\right)\mathbf{Q}\right]\mathbf{y}$$

$$\simeq \frac{N}{n}\frac{1}{\hat{n}}\begin{bmatrix} \frac{\frac{1}{n}\operatorname{tr}\mathbf{K}_{\cos}(\hat{\mathbf{X}},\hat{\mathbf{X}})}{(1+\delta_{\cos})^2} & \frac{\frac{1}{n}\operatorname{tr}\frac{1}{n}\operatorname{tr}\mathbf{K}_{\sin}(\hat{\mathbf{X}},\hat{\mathbf{X}})}{(1+\delta_{\sin})^2} \end{bmatrix}\mathbf{\Omega}\begin{bmatrix} \mathbf{y}^\mathsf{T}\bar{\mathbf{Q}}\mathbf{K}_{\cos}\bar{\mathbf{Q}}\mathbf{y} \\ \mathbf{y}^\mathsf{T}\bar{\mathbf{Q}}\mathbf{K}_{\sin}\bar{\mathbf{Q}}\mathbf{y} \end{bmatrix}.$$

where we apply Lemma 4 and recall

$$\mathbf{K}_{\cos}(\hat{\mathbf{X}},\hat{\mathbf{X}}) \equiv \left\{e^{-\frac{1}{2}(\|\hat{\mathbf{x}}_i\|^2 + \|\hat{\mathbf{x}}_j\|^2)}\cosh(\hat{\mathbf{x}}_i^\mathsf{T}\hat{\mathbf{x}}_j)\right\}_{i,j=1}^{\hat{n}}, \quad \mathbf{K}_{\sin}(\hat{\mathbf{X}},\hat{\mathbf{X}}) \equiv \left\{e^{-\frac{1}{2}(\|\hat{\mathbf{x}}_i\|^2 + \|\hat{\mathbf{x}}_j\|^2)}\sinh(\hat{\mathbf{x}}_i^\mathsf{T}\hat{\mathbf{x}}_j)\right\}_{i,j=1}^{\hat{n}}$$

Moving on to $\mathbf{Z}_2$ and we write

$$\mathbf{Z}_2 \equiv \frac{1}{n^2\hat{n}}\mathbb{E}\sum_{i=1}^N\sum_{j\neq i}\mathbf{y}^\mathsf{T}\mathbf{Q}\mathbf{U}_i\hat{\mathbf{U}}_i^\mathsf{T}\hat{\mathbf{U}}_j\mathbf{U}_j^\mathsf{T}\mathbf{Q}\mathbf{y}$$

$$= \frac{1}{n^2\hat{n}}\mathbb{E}\sum_{i=1}^N\sum_{j\neq i}\mathbf{y}^\mathsf{T}\mathbf{Q}_{-j}\mathbf{U}_i\hat{\mathbf{U}}_i^\mathsf{T}\hat{\mathbf{U}}_j(\mathbf{I}_2 + \tfrac{1}{n}\mathbf{U}_j^\mathsf{T}\mathbf{Q}_{-j}\mathbf{U}_j)^{-1}\mathbf{U}_j^\mathsf{T}\mathbf{Q}_{-j}\mathbf{y}$$

$$- \frac{1}{n^2\hat{n}}\mathbb{E}\sum_{i=1}^N\sum_{j\neq i}\mathbf{y}^\mathsf{T}\mathbf{Q}_{-j}\mathbf{U}_j(\mathbf{I}_2 + \tfrac{1}{n}\mathbf{U}_j^\mathsf{T}\mathbf{Q}_{-j}\mathbf{U}_j)^{-1}\mathbf{U}_j^\mathsf{T}\mathbf{Q}_{-j}\mathbf{U}_i\hat{\mathbf{U}}_i^\mathsf{T}\hat{\mathbf{U}}_j(\mathbf{I}_2 + \tfrac{1}{n}\mathbf{U}_j^\mathsf{T}\mathbf{Q}_{-j}\mathbf{U}_j)^{-1}\mathbf{U}_j^\mathsf{T}\mathbf{Q}_{-j}\mathbf{y}$$

$$\simeq \frac{1}{n\hat{n}}\mathbb{E}\sum_{i=1}^N\sum_{j\neq i}\mathbf{y}^\mathsf{T}\mathbf{Q}_{-j}\mathbf{U}_i\hat{\mathbf{U}}_i^\mathsf{T}\left(\frac{\mathbf{K}_{\cos}(\hat{\mathbf{X}},\mathbf{X})}{1+\delta_{\cos}} + \frac{\mathbf{K}_{\sin}(\hat{\mathbf{X}},\mathbf{X})}{1+\delta_{\sin}}\right)\mathbf{Q}_{-j}\mathbf{y}$$

$$- \frac{1}{n^2\hat{n}}\mathbb{E}\sum_{i=1}^N\sum_{j\neq i}\mathbf{y}^\mathsf{T}\mathbf{Q}_{-j}\mathbf{U}_j\begin{bmatrix} \frac{1}{1+\delta_{\cos}} & 0 \\ 0 & \frac{1}{1+\delta_{\sin}} \end{bmatrix}\begin{bmatrix} \frac{1}{n}\operatorname{tr}(\mathbf{Q}_{-j}\mathbf{U}_i\hat{\mathbf{U}}_i^\mathsf{T}\mathbf{K}_{\cos}(\hat{\mathbf{X}},\mathbf{X})) & 0 \\ 0 & \frac{1}{n}\operatorname{tr}(\mathbf{Q}_{-j}\mathbf{U}_i\hat{\mathbf{U}}_i^\mathsf{T}\mathbf{K}_{\sin}(\hat{\mathbf{X}},\mathbf{X})) \end{bmatrix}$$

$$\begin{bmatrix} \frac{1}{1+\delta_{\cos}} & 0 \\ 0 & \frac{1}{1+\delta_{\sin}} \end{bmatrix}\mathbf{U}_j^\mathsf{T}\mathbf{Q}_{-j}\mathbf{y} \equiv \mathbf{Z}_{21} - \mathbf{Z}_{22}.$$

For the term $\mathbf{Z}_{21}$, note that $\mathbf{Q}_{-j} \simeq \mathbf{Q}$ and **depends** on $\mathbf{U}_i$ (and $\hat{\mathbf{U}}_i$), such that

$$\mathbf{Z}_{21} \equiv \frac{1}{n^2 \hat{n}} \mathbb{E} \sum_{i=1}^{N} \sum_{j \neq i} \mathbf{y}^\mathsf{T} \mathbf{Q}_{-j} \mathbf{U}_i \hat{\mathbf{U}}_i^\mathsf{T} \left( \frac{\mathbf{K}_{\cos}(\hat{\mathbf{X}}, \mathbf{X})}{1 + \delta_{\cos}} + \frac{\mathbf{K}_{\sin}(\hat{\mathbf{X}}, \mathbf{X})}{1 + \delta_{\sin}} \right) \mathbf{Q}_{-j} \mathbf{y}$$

$$\simeq \frac{N}{n} \frac{1}{n\hat{n}} \mathbb{E} \sum_{i=1}^{N} \mathbf{y}^\mathsf{T} \mathbf{Q} \mathbf{U}_i \hat{\mathbf{U}}_i^\mathsf{T} \left( \frac{\mathbf{K}_{\cos}(\hat{\mathbf{X}}, \mathbf{X})}{1 + \delta_{\cos}} + \frac{\mathbf{K}_{\sin}(\hat{\mathbf{X}}, \mathbf{X})}{1 + \delta_{\sin}} \right) \mathbf{Q} \mathbf{y}$$

$$= \frac{N}{n} \frac{1}{n\hat{n}} \mathbb{E} \sum_{i=1}^{N} \mathbf{y}^\mathsf{T} \mathbf{Q}_{-i} \mathbf{U}_i (\mathbf{I}_2 + \frac{1}{n} \mathbf{U}_i^\mathsf{T} \mathbf{Q}_{-i} \mathbf{U}_i)^{-1} \hat{\mathbf{U}}_i^\mathsf{T} \hat{\boldsymbol{\Phi}} \mathbf{Q}_{-i} \mathbf{y}$$

$$- \frac{N}{n} \frac{1}{n\hat{n}} \mathbb{E} \sum_{i=1}^{N} \mathbf{y}^\mathsf{T} \mathbf{Q}_{-i} \mathbf{U}_i (\mathbf{I}_2 + \frac{1}{n} \mathbf{U}_i^\mathsf{T} \mathbf{Q}_{-i} \mathbf{U}_i)^{-1} \hat{\mathbf{U}}_i^\mathsf{T} \hat{\boldsymbol{\Phi}} \mathbf{Q}_{-i} \mathbf{U}_i (\mathbf{I}_2 + \frac{1}{n} \mathbf{U}_i^\mathsf{T} \mathbf{Q}_{-i} \mathbf{U}_i)^{-1} \mathbf{U}_i^\mathsf{T} \mathbf{Q}_{-i} \mathbf{y}$$

$$\simeq \frac{N}{n} \frac{1}{n\hat{n}} \mathbb{E} \sum_{i=1}^{N} \mathbf{y}^\mathsf{T} \mathbf{Q}_{-i} \left( \frac{\mathbf{K}_{\cos}(\hat{\mathbf{X}}, \mathbf{X})}{1 + \delta_{\cos}} + \frac{\mathbf{K}_{\sin}(\hat{\mathbf{X}}, \mathbf{X})}{1 + \delta_{\sin}} \right)^\mathsf{T} \hat{\boldsymbol{\Phi}} \mathbf{Q}_{-i} \mathbf{y}$$

$$- \frac{N}{n} \frac{1}{\hat{n}} \mathbb{E} \sum_{i=1}^{N} \mathbf{y}^\mathsf{T} \mathbf{Q}_{-i} \mathbf{U}_i \begin{bmatrix} \frac{1}{1+\delta_{\cos}} & 0 \\ 0 & \frac{1}{1+\delta_{\sin}} \end{bmatrix} \frac{1}{n} \hat{\mathbf{U}}_i^\mathsf{T} \hat{\boldsymbol{\Phi}} \mathbf{Q}_{-i} \mathbf{U}_i \begin{bmatrix} \frac{1}{1+\delta_{\cos}} & 0 \\ 0 & \frac{1}{1+\delta_{\sin}} \end{bmatrix} \mathbf{U}_i^\mathsf{T} \mathbf{Q}_{-i} \mathbf{y}$$

where we recall the shortcut $\boldsymbol{\Phi} \equiv \frac{\mathbf{K}_{\cos}}{1+\delta_{\cos}} + \frac{\mathbf{K}_{\sin}}{1+\delta_{\sin}}$ and similarly $\hat{\boldsymbol{\Phi}} \equiv \frac{\mathbf{K}_{\cos}(\hat{\mathbf{X}}, \mathbf{X})}{1+\delta_{\cos}} + \frac{\mathbf{K}_{\sin}(\hat{\mathbf{X}}, \mathbf{X})}{1+\delta_{\sin}} \in \mathbb{R}^{\hat{n} \times n}$.
As a consequence, we further have, with Lemma 4 that

$$\mathbf{Z}_{21} \simeq \left( \frac{N}{n} \right)^2 \frac{1}{\hat{n}} \mathbf{y}^\mathsf{T} \mathbb{E} \left[ \mathbf{Q} \hat{\boldsymbol{\Phi}}^\mathsf{T} \hat{\boldsymbol{\Phi}} \mathbf{Q} \right] \mathbf{y}$$

$$- \frac{N}{n} \frac{1}{\hat{n}} \mathbb{E} \sum_{i=1}^{N} \mathbf{y}^\mathsf{T} \mathbf{Q}_{-i} \mathbf{U}_i \begin{bmatrix} \frac{1}{1+\delta_{\cos}} & 0 \\ 0 & \frac{1}{1+\delta_{\sin}} \end{bmatrix} \begin{bmatrix} \frac{1}{n} \operatorname{tr}(\hat{\boldsymbol{\Phi}} \bar{\mathbf{Q}} \mathbf{K}_{\cos}(\hat{\mathbf{X}}, \mathbf{X})^\mathsf{T}) & 0 \\ 0 & \frac{1}{n} \operatorname{tr}(\hat{\boldsymbol{\Phi}} \bar{\mathbf{Q}} \mathbf{K}_{\sin}(\hat{\mathbf{X}}, \mathbf{X})^\mathsf{T}) \end{bmatrix}$$

$$\times \begin{bmatrix} \frac{1}{1+\delta_{\cos}} & 0 \\ 0 & \frac{1}{1+\delta_{\sin}} \end{bmatrix} \mathbf{U}_i^\mathsf{T} \mathbf{Q}_{-i} \mathbf{y}$$

$$\simeq \left( \frac{N}{n} \right)^2 \frac{1}{\hat{n}} \mathbf{y}^\mathsf{T} \mathbb{E} \left[ \mathbf{Q} \hat{\boldsymbol{\Phi}}^\mathsf{T} \hat{\boldsymbol{\Phi}} \mathbf{Q} \right] \mathbf{y}$$

$$- \left( \frac{N}{n} \right)^2 \frac{1}{\hat{n}} \mathbb{E} \mathbf{y}^\mathsf{T} \mathbf{Q} \left( \frac{1}{n} \operatorname{tr}(\hat{\boldsymbol{\Phi}} \bar{\mathbf{Q}} \mathbf{K}_{\cos}(\hat{\mathbf{X}}, \mathbf{X})^\mathsf{T}) \frac{\mathbf{K}_{\cos}}{(1 + \delta_{\cos})^2} + \frac{1}{n} \operatorname{tr}(\hat{\boldsymbol{\Phi}} \bar{\mathbf{Q}} \mathbf{K}_{\sin}(\hat{\mathbf{X}}, \mathbf{X})^\mathsf{T}) \frac{\mathbf{K}_{\sin}}{(1 + \delta_{\sin})^2} \right) \mathbf{Q} \mathbf{y}$$

$$\simeq \left( \frac{N}{n} \right)^2 \frac{1}{\hat{n}} \mathbf{y}^\mathsf{T} \mathbb{E} \left[ \mathbf{Q} \hat{\boldsymbol{\Phi}}^\mathsf{T} \hat{\boldsymbol{\Phi}} \mathbf{Q} \right] \mathbf{y} - \left( \frac{N}{n} \right)^2 \frac{1}{\hat{n}} \mathbf{y}^\mathsf{T} \left( \begin{bmatrix} \frac{\frac{1}{n} \operatorname{tr}(\hat{\boldsymbol{\Phi}} \bar{\mathbf{Q}} \mathbf{K}_{\cos}(\hat{\mathbf{X}}, \mathbf{X})^\mathsf{T})}{(1+\delta_{\cos})^2} & \frac{\frac{1}{n} \operatorname{tr}(\hat{\boldsymbol{\Phi}} \bar{\mathbf{Q}} \mathbf{K}_{\sin}(\hat{\mathbf{X}}, \mathbf{X})^\mathsf{T})}{(1+\delta_{\sin})^2} \end{bmatrix} \mathbb{E} \begin{bmatrix} \mathbf{Q} \mathbf{K}_{\cos} \mathbf{Q} \\ \mathbf{Q} \mathbf{K}_{\sin} \mathbf{Q} \end{bmatrix} \right) \mathbf{y}$$

$$\simeq \left( \frac{N}{n} \right)^2 \frac{1}{\hat{n}} \mathbf{y}^\mathsf{T} \bar{\mathbf{Q}} \boldsymbol{\Phi}^\mathsf{T} \hat{\boldsymbol{\Phi}} \bar{\mathbf{Q}} \mathbf{y}$$

$$+ \left( \frac{N}{n} \right)^2 \frac{1}{\hat{n}} \begin{bmatrix} \frac{\frac{1}{n} \operatorname{tr} \bar{\mathbf{Q}} \frac{N}{n} \hat{\boldsymbol{\Phi}}^\mathsf{T} \hat{\boldsymbol{\Phi}} \bar{\mathbf{Q}} \mathbf{K}_{\cos} - \frac{1}{n} \operatorname{tr} \bar{\mathbf{Q}} \hat{\boldsymbol{\Phi}} \mathbf{K}_{\cos}(\hat{\mathbf{X}}, \mathbf{X})}{(1+\delta_{\cos})^2} & \frac{\frac{1}{n} \operatorname{tr} \bar{\mathbf{Q}} \frac{N}{n} \hat{\boldsymbol{\Phi}}^\mathsf{T} \hat{\boldsymbol{\Phi}} \bar{\mathbf{Q}} \mathbf{K}_{\sin} - \frac{1}{n} \operatorname{tr} \bar{\mathbf{Q}} \hat{\boldsymbol{\Phi}}^\mathsf{T} \mathbf{K}_{\sin}(\hat{\mathbf{X}}, \mathbf{X})}{(1+\delta_{\sin})^2} \end{bmatrix} \boldsymbol{\Omega} \begin{bmatrix} \mathbf{y}^\mathsf{T} \bar{\mathbf{Q}} \mathbf{K}_{\cos} \bar{\mathbf{Q}} \mathbf{y} \\ \mathbf{y}^\mathsf{T} \bar{\mathbf{Q}} \mathbf{K}_{\sin} \bar{\mathbf{Q}} \mathbf{y} \end{bmatrix}$$

The last term $\mathbf{Z}_{22}$ can be similarly treated as

$$\mathbf{Z}_{22} \simeq \frac{1}{n^2 \hat{n}} \mathbb{E} \sum_{i=1}^{N} \sum_{j \neq i} \mathbf{y}^\mathsf{T} \mathbf{Q}_{-j} \mathbf{U}_j \begin{bmatrix} \frac{\frac{1}{n} \operatorname{tr}(\mathbf{Q} \mathbf{U}_i \hat{\mathbf{U}}_i^\mathsf{T} \mathbf{K}_{\cos}(\hat{\mathbf{X}}, \mathbf{X}))}{(1+\delta_{\cos})^2} & 0 \\ 0 & \frac{\frac{1}{n} \operatorname{tr}(\mathbf{Q} \mathbf{U}_i \hat{\mathbf{U}}_i^\mathsf{T} \mathbf{K}_{\sin}(\hat{\mathbf{X}}, \mathbf{X}))}{(1+\delta_{\sin})^2} \end{bmatrix} \mathbf{U}_j^\mathsf{T} \mathbf{Q}_{-j} \mathbf{y}$$

where by Lemma 2 we deduce

$$\frac{1}{n} \operatorname{tr}(\mathbf{Q} \mathbf{U}_i \hat{\mathbf{U}}_i^\mathsf{T} \mathbf{K}_{\cos}(\hat{\mathbf{X}}, \mathbf{X})) \simeq \frac{1}{n} \operatorname{tr} \left( \mathbf{Q}_{-i} \mathbf{U}_i (\mathbf{I}_2 + \mathbf{U}_i^\mathsf{T} \mathbf{Q}_{-i} \mathbf{U}_i)^{-1} \hat{\mathbf{U}}_i^\mathsf{T} \mathbf{K}_{\cos}(\hat{\mathbf{X}}, \mathbf{X}) \right)$$

$$\simeq \frac{1}{n} \operatorname{tr} \left( \mathbf{Q}_{-i} \mathbf{U}_i \begin{bmatrix} \frac{1}{1+\delta_{\cos}} & 0 \\ 0 & \frac{1}{1+\delta_{\sin}} \end{bmatrix} \hat{\mathbf{U}}_i^\mathsf{T} \mathbf{K}_{\cos}(\hat{\mathbf{X}}, \mathbf{X}) \right) \simeq \frac{1}{n} \operatorname{tr}(\bar{\mathbf{Q}} \hat{\boldsymbol{\Phi}}^\mathsf{T} \mathbf{K}_{\cos}(\hat{\mathbf{X}}, \mathbf{X}))$$

so that by again Lemma 4

$$\mathbf{Z}_{22} \simeq \frac{N}{n} \frac{1}{n\hat{n}} \mathbb{E} \sum_{j=1}^{N} \mathbf{y}^{\mathsf{T}} \mathbf{Q}_{-j} \mathbf{U}_j \begin{bmatrix} \frac{\frac{1}{n}\operatorname{tr}(\bar{\mathbf{Q}}\hat{\boldsymbol{\Phi}}^{\mathsf{T}}\mathbf{K}_{\cos}(\hat{\mathbf{X}},\mathbf{X}))}{(1+\delta_{\cos})^2} & 0 \\ 0 & \frac{\frac{1}{n}\operatorname{tr}(\bar{\mathbf{Q}}\hat{\boldsymbol{\Phi}}^{\mathsf{T}}\mathbf{K}_{\sin}(\hat{\mathbf{X}},\mathbf{X}))}{(1+\delta_{\sin})^2} \end{bmatrix} \mathbf{U}_j^{\mathsf{T}} \mathbf{Q}_{-j} \mathbf{y}$$

$$\simeq \left(\frac{N}{n}\right)^2 \frac{1}{\hat{n}} \mathbf{y}^{\mathsf{T}} \mathbb{E} \left[ \mathbf{Q} \left( \frac{\frac{1}{n}\operatorname{tr}(\bar{\mathbf{Q}}\hat{\boldsymbol{\Phi}}^{\mathsf{T}}\mathbf{K}_{\cos}(\hat{\mathbf{X}},\mathbf{X}))}{(1+\delta_{\cos})^2} \mathbf{K}_{\cos} + \frac{\frac{1}{n}\operatorname{tr}(\bar{\mathbf{Q}}\hat{\boldsymbol{\Phi}}^{\mathsf{T}}\mathbf{K}_{\sin}(\hat{\mathbf{X}},\mathbf{X}))}{(1+\delta_{\sin})^2} \mathbf{K}_{\sin} \right) \mathbf{Q} \right] \mathbf{y}$$

$$\simeq \left(\frac{N}{n}\right)^2 \frac{1}{\hat{n}} \mathbf{y}^{\mathsf{T}} \left( \bar{\mathbf{Q}}\boldsymbol{\Xi}\bar{\mathbf{Q}} + \frac{N}{n} \begin{bmatrix} \frac{\frac{1}{n}\operatorname{tr}(\bar{\mathbf{Q}}\boldsymbol{\Xi}\bar{\mathbf{Q}}\mathbf{K}_{\cos})}{(1+\delta_{\cos})^2} & \frac{\frac{1}{n}\operatorname{tr}(\bar{\mathbf{Q}}\boldsymbol{\Xi}\bar{\mathbf{Q}}\mathbf{K}_{\sin})}{(1+\delta_{\sin})^2} \end{bmatrix} \boldsymbol{\Omega} \begin{bmatrix} \bar{\mathbf{Q}}\mathbf{K}_{\cos}\bar{\mathbf{Q}} \\ \bar{\mathbf{Q}}\mathbf{K}_{\sin}\bar{\mathbf{Q}} \end{bmatrix} \right) \mathbf{y}$$

$$\simeq \left(\frac{N}{n}\right)^2 \frac{1}{\hat{n}} \begin{bmatrix} \frac{\frac{1}{n}\operatorname{tr}(\bar{\mathbf{Q}}\hat{\boldsymbol{\Phi}}^{\mathsf{T}}\mathbf{K}_{\cos}(\hat{\mathbf{X}},\mathbf{X}))}{(1+\delta_{\cos})^2} & \frac{\frac{1}{n}\operatorname{tr}(\bar{\mathbf{Q}}\hat{\boldsymbol{\Phi}}^{\mathsf{T}}\mathbf{K}_{\sin}(\hat{\mathbf{X}},\mathbf{X}))}{(1+\delta_{\sin})^2} \end{bmatrix} \boldsymbol{\Omega} \begin{bmatrix} \mathbf{y}^{\mathsf{T}}\bar{\mathbf{Q}}\mathbf{K}_{\cos}\bar{\mathbf{Q}}\mathbf{y} \\ \mathbf{y}^{\mathsf{T}}\bar{\mathbf{Q}}\mathbf{K}_{\sin}\bar{\mathbf{Q}}\mathbf{y} \end{bmatrix}.$$

Assembling the estimates for $\mathbf{Z}_1$, $\mathbf{Z}_{21}$ and $\mathbf{Z}_{22}$, we get

$$\mathbb{E}[E_{\text{test}}] \simeq \frac{1}{\hat{n}} \|\hat{\mathbf{y}}\|^2 - \frac{2}{\hat{n}} \hat{\mathbf{y}}^{\mathsf{T}} \frac{N}{n} \hat{\boldsymbol{\Phi}}\bar{\mathbf{Q}}\mathbf{y} + \frac{1}{\hat{n}} \mathbf{y}^{\mathsf{T}} \left( \frac{N^2}{n^2} \mathbf{Q}\hat{\boldsymbol{\Phi}}^{\mathsf{T}}\hat{\boldsymbol{\Phi}}\bar{\mathbf{Q}} \right) \mathbf{y} + \left(\frac{N}{n}\right)^2 \frac{1}{n\hat{n}} \times$$

$$\begin{bmatrix} \frac{\frac{n}{N}\operatorname{tr}\mathbf{K}_{\cos}(\hat{\mathbf{X}},\hat{\mathbf{X}}) + \frac{N}{n}\operatorname{tr}\bar{\mathbf{Q}}\hat{\boldsymbol{\Phi}}^{\mathsf{T}}\hat{\boldsymbol{\Phi}}\bar{\mathbf{Q}}\mathbf{K}_{\cos} - 2\operatorname{tr}\bar{\mathbf{Q}}\hat{\boldsymbol{\Phi}}^{\mathsf{T}}\mathbf{K}_{\cos}(\hat{\mathbf{X}},\mathbf{X})}{(1+\delta_{\cos})^2} & \frac{\frac{n}{N}\operatorname{tr}\mathbf{K}_{\sin}(\hat{\mathbf{X}},\hat{\mathbf{X}}) + \frac{N}{n}\operatorname{tr}\bar{\mathbf{Q}}\hat{\boldsymbol{\Phi}}^{\mathsf{T}}\hat{\boldsymbol{\Phi}}\bar{\mathbf{Q}}\mathbf{K}_{\sin} - 2\bar{\mathbf{Q}}\hat{\boldsymbol{\Phi}}^{\mathsf{T}}\mathbf{K}_{\sin}(\hat{\mathbf{X}},\mathbf{X})}{(1+\delta_{\sin})^2} \end{bmatrix}$$

$$\times \boldsymbol{\Omega} \begin{bmatrix} \mathbf{y}^{\mathsf{T}}\bar{\mathbf{Q}}\mathbf{K}_{\cos}\bar{\mathbf{Q}}\mathbf{y} \\ \mathbf{y}^{\mathsf{T}}\bar{\mathbf{Q}}\mathbf{K}_{\sin}\bar{\mathbf{Q}}\mathbf{y} \end{bmatrix}$$

which, up to further simplifications, concludes the proof of Theorem 3.

# E  Several Useful Lemmas

**Lemma 5** (Some useful properties of $\boldsymbol{\Omega}$)**.** *For any $\lambda > 0$ and $\boldsymbol{\Omega}$ defined in (8), we have*

1. *all entries of $\boldsymbol{\Omega}$ are positive;*

2. *for $2N = n$, $\det(\boldsymbol{\Omega}^{-1})$, as well as the entries of $\boldsymbol{\Omega}$, scales like $\lambda$ as $\lambda \to 0$;*

*Proof.* Developing the inverse we obtain

$$\boldsymbol{\Omega} = \begin{bmatrix} 1 - \frac{N}{n} \frac{\frac{1}{n}\operatorname{tr}(\bar{\mathbf{Q}}\mathbf{K}_{\cos}\bar{\mathbf{Q}}\mathbf{K}_{\cos})}{(1+\delta_{\cos})^2} & -\frac{N}{n} \frac{\frac{1}{n}\operatorname{tr}(\bar{\mathbf{Q}}\mathbf{K}_{\cos}\bar{\mathbf{Q}}\mathbf{K}_{\sin})}{(1+\delta_{\sin})^2} \\ -\frac{N}{n} \frac{\frac{1}{n}\operatorname{tr}(\bar{\mathbf{Q}}\mathbf{K}_{\cos}\bar{\mathbf{Q}}\mathbf{K}_{\sin})}{(1+\delta_{\cos})^2} & 1 - \frac{N}{n} \frac{\frac{1}{n}\operatorname{tr}(\bar{\mathbf{Q}}\mathbf{K}_{\sin}\bar{\mathbf{Q}}\mathbf{K}_{\sin})}{(1+\delta_{\sin})^2} \end{bmatrix}^{-1}$$

we have $[\boldsymbol{\Omega}^{-1}]_{11} = \frac{1}{1+\delta_{\cos}} + \frac{\lambda}{n}\operatorname{tr}\bar{\mathbf{Q}}\frac{\mathbf{K}_{\cos}}{1+\delta_{\cos}}\bar{\mathbf{Q}} + \frac{N}{n}\frac{1}{n}\operatorname{tr}\bar{\mathbf{Q}}\frac{\mathbf{K}_{\cos}}{1+\delta_{\cos}}\bar{\mathbf{Q}}\frac{\mathbf{K}_{\sin}}{1+\delta_{\sin}} > 0$, $[\boldsymbol{\Omega}^{-1}]_{12} < 0$, and similarly $[\boldsymbol{\Omega}^{-1}]_{21} < 0$, $[\boldsymbol{\Omega}^{-1}]_{22} > 0$. Furthermore, the determinant writes

$$\det(\boldsymbol{\Omega}^{-1}) = \left( 1 - \frac{1}{n}\operatorname{tr}\bar{\mathbf{Q}}\frac{\mathbf{K}_{\cos}}{1+\delta_{\cos}} + \frac{\lambda}{n}\operatorname{tr}\bar{\mathbf{Q}}\frac{\mathbf{K}_{\cos}}{1+\delta_{\cos}}\bar{\mathbf{Q}} \right) \left( 1 - \frac{1}{n}\operatorname{tr}\bar{\mathbf{Q}}\frac{\mathbf{K}_{\sin}}{1+\delta_{\sin}} + \frac{\lambda}{n}\operatorname{tr}\bar{\mathbf{Q}}\frac{\mathbf{K}_{\sin}}{1+\delta_{\sin}}\bar{\mathbf{Q}} \right)$$

$$+ \left( 1 - \frac{1}{n}\operatorname{tr}\bar{\mathbf{Q}}\frac{\mathbf{K}_{\cos}}{1+\delta_{\cos}} + 1 - \frac{1}{n}\operatorname{tr}\bar{\mathbf{Q}}\frac{\mathbf{K}_{\sin}}{1+\delta_{\sin}} + \frac{\lambda}{n}\operatorname{tr}\bar{\mathbf{Q}}\left( \frac{\mathbf{K}_{\cos}}{1+\delta_{\cos}} + \frac{\mathbf{K}_{\sin}}{1+\delta_{\sin}} \right)\bar{\mathbf{Q}} \right)$$

$$\times \frac{N}{n}\frac{1}{n}\operatorname{tr}\bar{\mathbf{Q}}\frac{\mathbf{K}_{\cos}}{1+\delta_{\cos}}\bar{\mathbf{Q}}\frac{\mathbf{K}_{\sin}}{1+\delta_{\sin}}$$

where we constantly use the fact that $\bar{\mathbf{Q}}\frac{N}{n}\left( \frac{\mathbf{K}_{\cos}}{1+\delta_{\cos}} + \frac{\mathbf{K}_{\sin}}{1+\delta_{\sin}} \right) = \mathbf{I}_n - \lambda\bar{\mathbf{Q}}$. Note that

$$1 - \frac{1}{n}\operatorname{tr}\bar{\mathbf{Q}}\frac{\mathbf{K}_{\cos}}{1+\delta_{\cos}} = \frac{1}{1+\delta_{\cos}} > 0, \quad 1 - \frac{1}{n}\operatorname{tr}\bar{\mathbf{Q}}\frac{\mathbf{K}_{\sin}}{1+\delta_{\sin}} = \frac{1}{1+\delta_{\sin}} > 0$$

$$\frac{1}{1+\delta_{\cos}} + \frac{1}{1+\delta_{\sin}} = 2 - \underline{\frac{n}{N}} + \frac{\lambda}{N}\operatorname{tr}\bar{\mathbf{Q}} > 0$$

so that 1) $\det(\boldsymbol{\Omega}^{-1}) > 0$ and 2) for $2N = n$, $\det(\boldsymbol{\Omega}^{-1})$ scales like $\lambda$ as $\lambda \to 0$. $\qquad\square$

**Lemma 6** (Derivatives with respect to $N$). *Let Assumption 1 holds, for any $\lambda > 0$ and*

$$
\begin{cases}
\delta_{\cos} = \frac{1}{n} \operatorname{tr}(\mathbf{K}_{\cos}\bar{\mathbf{Q}}) = \frac{1}{n} \operatorname{tr} \mathbf{K}_{\cos} \left( \frac{N}{n} \left( \frac{\mathbf{K}_{\cos}}{1+\delta_{\cos}} + \frac{\mathbf{K}_{\sin}}{1+\delta_{\sin}} \right) + \lambda \mathbf{I}_n \right)^{-1} \\
\delta_{\sin} = \frac{1}{n} \operatorname{tr}(\mathbf{K}_{\sin}\bar{\mathbf{Q}}) = \frac{1}{n} \operatorname{tr} \mathbf{K}_{\sin} \left( \frac{N}{n} \left( \frac{\mathbf{K}_{\cos}}{1+\delta_{\cos}} + \frac{\mathbf{K}_{\sin}}{1+\delta_{\sin}} \right) + \lambda \mathbf{I}_n \right)^{-1}
\end{cases}
$$

*defined in Theorem 1, we have that $(\delta_{\cos}, \delta_{\sin})$ and $\|\bar{\mathbf{Q}}\|$ are all decreasing functions of $N$. Note in particular that the same conclusion holds for $2N > n$ as $\lambda \to 0$.*

*Proof.* We write

$$
\begin{bmatrix} \frac{\partial \delta_{\cos}}{\partial N} \\ \frac{\partial \delta_{\sin}}{\partial N} \end{bmatrix} = -\frac{1}{n}\mathbf{\Omega} \begin{bmatrix} \frac{1}{n} \operatorname{tr}\left(\bar{\mathbf{Q}}\mathbf{\Phi}\bar{\mathbf{Q}}\mathbf{K}_{\cos}\right) \\ \frac{1}{n} \operatorname{tr}\left(\bar{\mathbf{Q}}\mathbf{\Phi}\bar{\mathbf{Q}}\mathbf{K}_{\sin}\right) \end{bmatrix} = -\frac{n}{N}\frac{1}{n}\mathbf{\Omega} \begin{bmatrix} \delta_{\cos} - \frac{\lambda}{n} \operatorname{tr}(\bar{\mathbf{Q}}\mathbf{K}_{\cos}\bar{\mathbf{Q}}) \\ \delta_{\sin} - \frac{\lambda}{n} \operatorname{tr}(\bar{\mathbf{Q}}\mathbf{K}_{\sin}\bar{\mathbf{Q}}) \end{bmatrix} \tag{23}
$$

for $\mathbf{\Omega}$ defined in (8) and $\mathbf{\Phi} = \frac{\mathbf{K}_{\cos}}{1+\delta_{\cos}} + \frac{\mathbf{K}_{\sin}}{1+\delta_{\sin}}$, which, together with Lemma 5, allows us to conclude that $\frac{\partial \delta_{\cos}}{\partial N}, \frac{\partial \delta_{\sin}}{\partial N} < 0$. Further note that

$$
\frac{\partial \bar{\mathbf{Q}}}{\partial N} = -\frac{1}{n}\bar{\mathbf{Q}} \left( \mathbf{\Phi} - \frac{\mathbf{K}_{\cos}}{(1+\delta_{\cos})^2} N \frac{\partial \delta_{\cos}}{\partial N} - \frac{\mathbf{K}_{\sin}}{(1+\delta_{\sin})^2} N \frac{\partial \delta_{\sin}}{\partial N} \right) \bar{\mathbf{Q}}
$$

which concludes the proof. $\qquad\square$

**Lemma 7** (Derivative with respect to $\lambda$). *For any $\lambda > 0$, $(\delta_{\cos}, \delta_{\sin})$ and $\|\bar{\mathbf{Q}}\|$ defined in Theorem 1 decrease as $\lambda$ grows large.*

*Proof.* Taking the derivative of $(\delta_{\cos}, \delta_{\sin})$ with respect to $\lambda > 0$, we have explicitly

$$
\begin{bmatrix} \frac{\partial \delta_{\cos}}{\partial \lambda} \\ \frac{\partial \delta_{\sin}}{\partial \lambda} \end{bmatrix} = -\mathbf{\Omega} \begin{bmatrix} \frac{1}{n} \operatorname{tr}(\bar{\mathbf{Q}}\mathbf{K}_{\cos}\bar{\mathbf{Q}}) \\ \frac{1}{n} \operatorname{tr}(\bar{\mathbf{Q}}\mathbf{K}_{\sin}\bar{\mathbf{Q}}) \end{bmatrix} \tag{24}
$$

which, together with the fact that all entries of $\mathbf{\Omega}$ are positive (Lemma 5), allows us to conclude that $\frac{\partial \delta_{\cos}}{\partial \lambda}, \frac{\partial \delta_{\sin}}{\partial \lambda} < 0$. Further considering

$$
\frac{\partial \bar{\mathbf{Q}}}{\partial \lambda} = \bar{\mathbf{Q}} \left( \frac{N}{n} \frac{\mathbf{K}_{\cos}}{(1+\delta_{\cos})^2} \frac{\partial \delta_{\cos}}{\partial \lambda} + \frac{N}{n} \frac{\mathbf{K}_{\sin}}{(1+\delta_{\sin})^2} \frac{\partial \delta_{\sin}}{\partial \lambda} - \mathbf{I}_n \right) \bar{\mathbf{Q}}
$$

and thus the conclusion for $\bar{\mathbf{Q}}$.

$\qquad\square$

# F   Additional Real-world Data sets

We have presented results in detail for one particular real-world data set, the MNIST data set, but we have extensive empirical results demonstrating that similar conclusions hold more broadly. As an example of this, here we present numerical evaluations of our results on several other real-world image data sets. We consider the classification task on another two MNIST-like data sets composed of $28 \times 28$ grayscale images: the Fashion-MNIST [53] and the Kannada-MNIST [42] data sets. Each image is represented as a $p = 784$-dimensional vector and the output targets $\mathbf{y}, \hat{\mathbf{y}}$ are taken to have $-1, +1$ entries depending on the image class. As a consequence, both the training and test MSEs defined in (4) are approximately 1 for $N = 0$ and not-too-large regularization $\lambda$, as observed in Figure 4 and Figure 10 below. For each data set, images were jointly centered and scaled so to fall close to the setting of Assumption 1 on $\mathbf{X}$ and $\hat{\mathbf{X}}$.

In Figure 8, we compare the empirical training and test NSEs with their limiting behaviors derived from Theorem 2 and 3, as a function of the penalty parameter $\lambda$, on a training set of size $n = 1\,024$ (512 images from class 5 and 512 images from class 6) with feature dimension $N = 256$, on both data sets. A close fit between theory and practice is observed, for moderately large values of $n, p, N$, demonstrating thus a wide practical applicability of the proposed asymptotic analyses, particularly compared to the (limiting) Gaussian kernel predictions per Figure 1.

In Figure 9, we report the behavior of the pair $(\delta_{\cos}, \delta_{\sin})$ for small values of $\lambda = 10^{-7}$ and $10^{-3}$. Similar to the two leftmost plots in Figure 3 for MNIST, a jump from the under- to over-parameterized

Figure 8: MSEs of RFF regression on Fashion-MNIST (**left two**) and Kannada-MNIST (**right two**) data (class 5 versus 6), as a function of regression parameter $\lambda$, for $p = 784$, $n = \hat{n} = 1\,024$, $N = 256$ and $512$. Empirical results displayed in **blue** (circles for training and crosses for test); and the asymptotics from Theorem 2 and 3 displayed in **red** (sold lines for training and dashed for test). Results obtained by averaging over 30 runs.

Figure 9: Behavior of $(\delta_{\cos}, \delta_{\sin})$ in (11), on Fashion-MNIST (**left two**) and Kannada-MNIST (**right two**) data (class 8 versus 9), for $p = 784$, $n = 1000$, $\lambda = 10^{-7}$ and $10^{-3}$. The **black** dashed line is the interpolation threshold $2N = n$.

regime occurs at the interpolation threshold $2N = n$, in both Fashion- and Kannada-MNIST data sets, clearly indicating the two phases of learning and the phase transition between them.

In Figure 10, we report the empirical and theoretical test errors as a function of the ratio $N/n$, on a training test of size $n = 500$ (250 images from class 8 and 250 images from class 9), by varying feature dimension $N$. An exceedingly small regularization $\lambda = 10^{-7}$ is applied to mimic the "ridgeless" limiting behavior as $\lambda \to 0$. On both data sets, double-descent-type test curves are observed where the test errors goes down and up, with a singular peak around $2N = n$, and then goes down monotonically as $N$ continues to increase when $2N > n$.

Figure 10: Empirical (crosses) and theoretical (dashed lines) test error of RFF regression, as a function of the ratio $N/n$, on Fashion-MNIST (**left two**) and Kannada-MNIST (**right two**) data (class 8 versus 9), for $p = 784$, $n = 500$, $\lambda = 10^{-7}$ and $10^{-3}$. The **black** dashed line is the interpolation threshold $2N = n$. Results obtained by averaging over 30 runs.