[Reviews · NeurIPS 2020]

Review 1

Summary and Contributions: The paper considers random Fourier features and analyzes the convergence of the empirical kernel matrix to its expectation. As opposed to previous work in which the number of random features is assumed to be significantly larger than the dimension and/or the number of examples, the current paper assumes that all these quantities are of the same order of magnitude. In this harder and often empirical regime, they prove that the empirical kernel converge to its expectation. This result is shown to have implication on the rate in which the test and train error decreases. Finally, the authors perform experiments that demonstrate that in practice, their bounds predict the true rates of convergence.

Strengths: The paper successfully analyses an interesting regime of parameters for the first time.

Weaknesses: I don't see a particular weakness

Correctness: yes

Clarity: yes

Relation to Prior Work: yes

Reproducibility: Yes

Additional Feedback:


Review 2

Summary and Contributions: This paper is a theoretical study of the training and test error for regression tasks using random Fourier features. The authors focus on the challenging limit where the number of samples n, their dimension p, and the number of random features N all go to infinity at the same rate. Their key result is a characterisation of the spectral density of the Gram matrix of the random features (Thm 1), which determines the test and training errors in the regression task (Thms 2 and 3). They obtain this result using methods recently introduced in random matrix theory (RMT) (see below), and their theory agrees really well with numerical simulations (Fig. 1). As an application, they re-write their exact result as a correction δ to the result one would obtain in the limit where only N→∞. Interestingly, these corrections undergo a phase transition as a function of N/n (Fig. 3) at low values of the regularisation parameter λ. This phase transition occurs at the interpolation threshold, where the system goes from an under- to an over-parameterised phase. This phase transition thus occurs exactly at the interpolation threshold, and has recently received a lot of attention in the context of the "double descent" phenomenon.

Strengths: Overall, this is a solid analysis of random feature regression that is quite close to earlier work by Mei and Montanari (1908.05355), who also studied random features regression in the same asymptotic limit by using tools from random matrix theory, and who also give an extensive discussion of the double descent. While Mei and Montanari consider inputs to be i.i.d. samples from the sphere, here the authors stress that their result holds "with almost no specific assumption on the data distribution". More specifically, their result on the training error (Thm 2) requires only a boundedness assumption on the data matrix, since the randomness needed to use RMT tools comes from the weights of the random features. For the test error (Thm 3), they assume that test data are behaving as concentrated random vectors (Assumption 2). This notion was recently introduced by Louart and Couillet (1805.08295; see also El Amine Seddik, Louart, Tamaazousti and Couillet, ICML '20).

Weaknesses: I find the paper a tad incremental with respect to Song Mei and Andrea Montanari's one, about one year ago. It was done for gaussian input whie here the authors consider much broader input distributions. However, this does not make any change for the training error, and it doesn't seem to change much for the test error once you assume that inputs are concentrated random vectors. Unfortunately, the author do not discuss concretely how their results, esp. for the test error, go beyond Mei & Montanari.

Correctness: The results are extremely solid random matrix theory. The present work extends the analysis of Mei and Montanari (1908.05355) of random feature regression for inputs drawn from the sphere to a wide range of input distributions.

Clarity: Yes

Relation to Prior Work: Yes

Reproducibility: Yes

Additional Feedback: ** Questions on the manuscript - Going from i.i.d. inputs to essentially any data matrix does not affect the result for the training error much, since that result exploits the randomness in the random features. Could you comment on how considering data generated from a GAN makes your result for the test error (Thm 3) different to the result of Mei and Montanari for i.i.d. inputs? ** Minor issues and typos - In Assumption 1, first condition, I found the notation confusing: it is not specified with respect to what the limit is taken, so I am assuming this is the limit n → ∞; how should I interpret infₙ then? - It would be nice if the authors clarified the norms they mean when they write || . ||. For example in Appendix A, is || . || the spectral norm? d to use RMT tools comes from the weights of the random features. For the test error (Thm 3), they assume that test data are behaving as concentrated random vectors (Assumption 2). This notion was recently introduced by Louart and Couillet (1805.08295; see also El Amine Seddik, Louart, Tamaazousti and Couillet, ICML '20). In summary, the present work extends the analysis of Mei and Montanari (1908.05355) of random feature regression for inputs drawn from the sphere to a wide range of input distributions. ** Questions on the manuscript - Going from i.i.d. inputs to essentially any data matrix does not affect the result for the training error much, since that result exploits the randomness in the random features. Could you comment on how considering data generated from a GAN makes your result for the test error (Thm 3) different to the result of Mei and Montanari for i.i.d. inputs? ** Minor issues and typos - In Assumption 1, first condition, I found the notation confusing: it is not specified with respect to what the limit is taken, so I am assuming this is the limit n → ∞; how should I interpret infₙ then? - It would be nice if the authors clarified the norms they mean when they write || . ||. For example in Appendix A, is || . || the spectral norm?


Review 3

Summary and Contributions: This paper studies random features ridge regression in a single-hidden-layer neural network model with random Fourier features, in the high-dimensional asymptotic limit where input dimension, sample size, and network width increase proportionally. It derives asymptotically equivalent expressions for the training and test errors that are deterministic conditional on the input data, and are valid under very general assumptions for the data. These are shown to match very closely the training and test errors observed on real data sets such as MNIST, Fashion-MNIST, and Kanada-MNIST. The asymptotically equivalent form of the resolvent matrix is quite clean, and highlights two "angle" parameters (delta_cos, delta_sin) which may be interpreted as responsible for interpolating between the behavior in the proportional-width regime of this work and the infinite-width regime as studied by e.g. Rahimi, Recht '08. The authors analyze the form of the asymptotic test error and argue that there are situations, when the test data differs enough from the training data, such that the test error exhibits a singularity in the ridgeless limit lambda -> 0 at the interpolation threshold 2N/n = 1. This connects to the recent interest in double-descent phenomena for the test error.

Strengths: I think this is a strong contribution to NeurIPS. There has been a flurry of work on studying random features regression in the high-dimensional setting considered in this paper, but most of the works I'm aware of derive asymptotically equivalent forms for the training and test errors that marginalize also over the distribution of training and test data (X,y). This requires assumptions for (X,y), e.g. i.i.d. Gaussian X and/or a specific form for the true function y = f(X). In contrast, the current work marginalizes only over the weights W, stating the results as functions of (X,y) and (X_test,y_test). The benefit of this perspective is that the results are almost assumption-free and are widely applicable, as evidenced by the agreement with the simulations on MNIST. The downside is that these results are in some sense only a "halfway-point" to the types of results that fully marginalize over (X,y), leading to forms for the training and test errors that still depend on high-dimensional inputs. Thus the burden is shifted somewhat to a more difficult analysis of these equivalent expressions for the training and test errors. But I think, at least in the specific model of Fourier features studied here, that the expressions obtained are sufficiently clean and simple. The authors make a good case in the paper that the expressions can be analyzed to yield insight on the qualitative behavior of the training and test errors, in more general contexts for the input data than has been achieved in previous analyses.

Weaknesses: I find the discussion around the test error weaker and a bit more difficult to understand than the discussion around the training error---see below for concrete comments/questions.

Correctness: Yes, I believe the claims are correct.

Clarity: Yes, the paper is well-written, and the results are well-explained and interpreted for a broad NeurIPS audience.

Relation to Prior Work: Yes, I believe this is adequately discussed.

Reproducibility: Yes

Additional Feedback: - I find the statement of Assumption 2 unclear: Are all training samples drawn from a single one of these K classes, or can different training samples come from different classes? Similarly, are all test samples drawn from a single class k, or can they come from different classes in {1,...,K}? If these can come from different classes, is there some assumption being made about how they are distributed across classes? - I'm also a little bit uncomfortable with how the assumption is used in the proof: It seems that it is used only to bound the operator norm of various matrix products involving the training resolvent Q and the test data matrix Sigma_{\hat{X}}? The steps that require such bounds are glossed over in the proof, and the argument for why Assumption 2 leads to such bounds is also omitted. Would it be possible to give a more general assumption that states the required bounds for Sigma_{\hat{X}}, point out more explicitly the steps in the proof where these types of bounds are used, and then maybe discuss that the condition in Assumption 2 is an example of when these bounds hold? If this latter discussion is basically the same as arguments of [Louart, Couillet '18], then perhaps a reference will suffice, but it'd be helpful to have the main result and assumption in a clearer form that does not require checking the detailed arguments of this reference. - There are aspects of the discussion on page 7 that are a bit confusing. On line 252 I'm not sure what \bar{Q} \sim \lambda^{-1} means, as it seems like there are some eigenspaces for which this scaling holds, and other eigenspaces for which \bar{Q} is bounded. This then pertains to parsing the expressions of Theorems 2 and 3 in Remark 3: For example are the quantities n^{-1} tr \bar{Q} K_cos \bar{Q} and y' \bar{Q} K_cos \bar{Q} y actually bounded, despite this blow-up in \bar{Q}? It's a little bit hard to tell the scalings of the various terms as lambda -> 0, and which terms are contributing to the divergent behavior. - The meaning of "test data \hat{X} is sufficiently different from the training data X" on line 260 is unclear to me. Are the arguments behind Remark 3 rigorous enough to be able to formalize this remark as a theorem? - There are maybe some omitted steps in the proof, in that the application of Lemma 3 on lines 485-487 shows concentration around quantities such as alpha_cos and alpha_sin defined with Q_{-i} in place of E[Q], so that one needs to bound the difference between these quantities and alpha_cos and alpha_sin as defined in eq. (14).


Review 4

Summary and Contributions: I believe the setup is like this: 1. There is an unknown function f: R^p -> R. 2. We are given training data. This will include X in R^{p x n} and Y in R^n, where Y_i is pretty close to f(X_1i,X_2i...X_pi) 4. We estimate fhat by ridge regressing Y against 2N random fourier features of X. 5. We will then be given some test data (Xhat, Yhat). This will include Xhat in R^{p x nhat} and Yhat in R^nhat, where Yhat_i is pretty close to f(Xhat_1i,Xhat_2i...Xhat_pi) 6. Our "error" will be measured as sum_i (fhat(Xhat_1i,...Xhat_pi)-Yhat_i)^2 The main theoretical contribution of the paper appears to be this. If... - the training covariates (X) aren't crazy different from the test covariates (Xhat) [[Assumption 2]] - n-> infinity - |log(n/p)| and |log(n/N)| stay bounded - the ridge regression is done by minimizing 1/n ||Y -Xfeatures^T beta||^2 + lambda||beta||^2 (with some fixed lambda) ...then we can get an interesting expression which approximates the test error (Theorem 3). The expression has two terms. - I believe the first term is large unless you could do a good job of guessing Yhat from Xhat using an interpolation-based regression trained on X,Y (e.g. k-nearest neighbors or something) - I don't understand the second term. The paper then uses this expression to try to learn something about how we should perform machine learning in practice.

Strengths: The work appears to be rigorous and the expressions in the Theorems are novel and interesting to stare at.

Weaknesses: Significance -- I could not figure out how this would actually help me think about how to regularize in machine learning or use RFF for a real-world regression problem. Post-author-feedback-edit: The authors point out that this paper shows it is hazardous to substitute the gram matrix for the kernel matrix. So I guess, in practice, random features might have worse test performance than you might have expected. Fair enough. But then you'd just look at the heldout performance and be like "huh, this isn't working. I should do something else." Basically I just can't imagine a situation where I'm trying to solve a problem, and this would effect my decisions about how to solve it. I think there is a nugget of wisdom in here somewhere, maybe something to do about how to think about regularization, but I just can't figure out what it is. But, as the authors point out, I may not be the real target audience of this paper! Thanks for the interesting discussion.

Correctness: The claims look plausible. In terms of empirical methodology: in the experiments section they look at error on heldout data. However, though mnist comes with its own test/train split, they do not use it. Instead, the testing data is constructed by adding some speckle noise to the training data. This doesn't really seem like testing data in any kind of real-world sense, and I'm not sure how to interpret it except in a very abstract sense.

Clarity: In making the analyses in this paper, I suspect that the authors have gained considerable insight into how machine learning should be done and how it should be regularized. However, I was unable to share in that insight by reading this article. For example, I'm not sure what I should actually think about Theorem 3? I suppose the most promising thing to focus on would be the double-descent. There must be some way of stripping out all of the many many parameters in Theorem 3 to arrive at a closed-form function of lambda which features the double descent. That would show how double descent pops out of those formulas. That would be interesting.

Relation to Prior Work: The connection to prior work appears to be reasonably complete.

Reproducibility: Yes

Additional Feedback: This is probably obvious for people in the field -- but the training loss is written lossA(beta;lambda) = 1/n ||Y -Xfeatures^T beta||^2 + lambda||beta||^2 I guess this is consistent with modern ML -- if you're training on minibatches of size K with a fixed L2 penalty of magnitude lambda*K, you're effectively doing something like this. Maybe. In a more classical stats world, I'd expect something like: lossB(beta;lambda) = ||Y -Xfeatures^T beta||^2 + lambda||beta||^2 Thus, in terms of the classical stats point of view, the regularization is getting scaled up with n in a very particular way. It feels like that must be pretty relevant to why we're seeing this double descent thing. Presumably there is an optimal rate at which to scale the regularization (given a fixed value for the ratio N/n). It feels like this paper should reveal the answer to this question. But I couldn't discover it.

[Author Response · NeurIPS 2020]

**Paper ID: 2097. Title: A random matrix analysis of random Fourier features: beyond the Gaussian kernel, a precise phase transition, and the corresponding double descent.**

We would like to thank the reviewers for their positive support and for their thorough and helpful remarks. The final submission of the paper will reflect their suggested revisions. The typographical errors will be fixed, and more details will be added to clarify the discussions and the proofs. Before addressing the reviewers' concerns individually, we wish to insist that, built upon previous efforts (e.g., of Mei and Montanari), one of the major objectives of this article is to extend the existing double descent analysis to a *more practical setting*. To this end, we proposed to analyze the popular random Fourier feature method, and to work with more generic data models. This allows us to conclude that double descent is *intrinsic* to random feature model and is *independent of the underlying data model*, as long as our mild technical assumptions are met. For the three most confident reviewers we focus on a few clarifying remarks. The fourth least confident reviewer had an anomalously-low score, and we focus on using his/her remarks to help clarify our main results for ML readers more generally.

**Reviewer #1**: We thank the reviewer for the positive support and constructive feedback.

**Reviewer #2**: Our result is a natural extension of the analysis of Mei and Montanari (1908.05355) and their results can be retrieved by taking data uniformly distributed on the unit sphere (which is a popular example of concentrated random vectors in our Assumption 2). By specifying the data and target model, Mei and Montanari reached more explicit results and established the double descent test curve. Our results hold for a much broader range of data models, and are thus of more practical interest. The proposed analysis, despite depending on the data kernel matrices, is still fully capable of characterizing the double descent phenomena and matches real-world experiments. More discussions will be made to better distinguish this work from previous efforts.

**Reviewer #3**: Assumption 2 does not impose any constraints on the number of training or test data in each class and is needed to bound the operator norms of matrices of the type $\mathbf{Q}\mathbf{\Sigma}_{\hat{\mathbf{X}}}^{\mathsf{T}}/\sqrt{n}$ and $\mathbf{Q}\mathbf{\Sigma}_{\mathbf{X}}^{\mathsf{T}}\mathbf{\Sigma}_{\hat{\mathbf{X}}}/n$. While we have the natural control $\|\mathbf{Q}\mathbf{\Sigma}_{\mathbf{X}}^{\mathsf{T}}/\sqrt{n}\|^2 \leq \|\mathbf{Q}\mathbf{\Sigma}_{\mathbf{X}}^{\mathsf{T}}\mathbf{\Sigma}_{\mathbf{X}}\mathbf{Q}/n\| \leq \lambda^{-1}$, it is in general not true under Assumption 1 if we replace $\mathbf{\Sigma}_{\mathbf{X}}$ by $\mathbf{\Sigma}_{\hat{\mathbf{X}}}$. This is needed in both $\mathbf{Z}_1$ and $\mathbf{Z}_2$ in the proof of Theorem 3 in Appendix D, for instance in the first approximation of $\mathbf{Z}_1$ to bound the difference when we replace $\mathbf{I}_2 + \frac{1}{n}\mathbf{U}_i^{\mathsf{T}}\mathbf{Q}_{-i}\mathbf{U}_i$ by its expectation (with respect to $\mathbf{W}$). More details will be added to the discussions and proofs to clarify more explicitly when and how Assumption 2 is used.

With respect to the divergent behavior of the test error as $\lambda \to 0$, it is indeed due to the two-by-two matrix $\mathbf{\Omega}$ (instead of $\bar{\mathbf{Q}}$) in Theorem 3 that scales like $\lambda^{-1}$ as $\lambda \to 0$ for $2N = n$. This is briefly discussed in Remark 3 and Section 3.3, with a proof in Lemma 5 of the appendix. We will state these results more explicitly in the final version of the submission.

**Reviewer #4**: We would like to clarify what appear to be several misunderstanding on the part of the review, and these constructive comments will be incorporated into the final version of this submission to help clarify the following issues.

Significance of this work: the theoretical analysis in the large $n, p, N$ regime (with in particular the number of random features $N$ not *much larger* than the sample size $n$) proposed in this work is, by itself, of considerable practical significance. While random feature techniques are proposed to alleviate the computational burden of large kernel matrices, and one thus expect to take $N < n$, our analysis shows in this practical $N \sim n \sim p$ regime that there is a significant mismatch between results obtained with the popular random Fourier feature and the "expected" Gaussian kernel. This is numerically supported by Figure 1 and theoretically explained by our Theorem 1-3. We thus argue that the simple substitution of the random Fourier Gram matrix by the Gaussian kernel matrix can be hazardous in most random feature-based methods, in the more practical $n \sim N$ regime.

Simpler characterization of double descent: as a consequence of the under- to over-parametrization phase transition behavior of the resolvent $\bar{\mathbf{Q}}$ discussed in Section 3.2, we observe in Remark 3 that the double descent test curve is a direct consequence of this phase transition and more precisely, of the singular behavior of the two-by-two matrix $\mathbf{\Omega}$ (that scales like $\lambda^{-1}$ as $\lambda \to 0$ at $2N = n$, with a proof in Lemma 5 of the appendix) in the second term of $\bar{E}_{\text{test}}$ in Theorem 3. We will state these results more explicitly in the final version of the submission.

With respect to the empirical results in Section 3, artificial noise is only added to the training data in Figure 5 of Section 3.3. There, the objective of adding Gaussian noise is to study, in a qualitative manner, the impact of training-and-test data similarity on the double descent test curve. More discussions will be added to better clarify this point.

We agree with the reviewer that, by taking the training loss to be $\frac{1}{n}\|\mathbf{y}-\mathbf{\Sigma}^{\mathsf{T}}\boldsymbol{\beta}\|^2+\lambda\|\boldsymbol{\beta}\|^2$ (instead of $\|\mathbf{y}-\mathbf{\Sigma}^{\mathsf{T}}\boldsymbol{\beta}\|^2+\lambda\|\boldsymbol{\beta}\|^2$ proposed by the reviewer), with $\lambda > 0$ and $\mathbf{\Sigma} = \sigma(\mathbf{W}\mathbf{X})$ the feature of data $\mathbf{X}$, we implicitly choose the scaling of the regularization $\lambda$. This scaling is chosen here so that the feature Gram matrix and the regularization are set "on even ground", in the sense that with, say, linear activation $\sigma(t) = t$ we have $\boldsymbol{\beta} = \frac{1}{n}\mathbf{W}\mathbf{X}(\frac{1}{n}\mathbf{X}^{\mathsf{T}}\mathbf{W}^{\mathsf{T}}\mathbf{W}\mathbf{X} + \lambda\mathbf{I}_n)^{-1}\mathbf{y}$, so that $\frac{1}{n}\mathbf{X}^{\mathsf{T}}\mathbf{W}^{\mathsf{T}}\mathbf{W}\mathbf{X}$ and $\lambda\mathbf{I}_n$ are both of operator norm of order $O(1)$, with standard Gaussian $\mathbf{W}$ under Assumption 1.

We thank the reviewers again for their time and help in improving our contribution.

[Meta-Review · NeurIPS 2020]

The reviewers and I agree that this paper provides a solid analysis of random Fourier features in the high-dimensional regime and should be accepted. While there was some discussion regarding the improvement over Mei and Montanari (2019), I am confident that the technical novelty of this paper is more than sufficient to merit publication.